# A KDM5–Prospero transcriptional axis functions during early neurodevelopment to regulate mushroom body formation

Hayden AM Hatch[1], Helen M Belalcazar[2], Owen J Marshall[3], Julie Secombe[1,2]*

[1]Dominick P. Purpura Department of Neuroscience Albert Einstein College of Medicine, Bronx, United States; [2]Department of Genetics Albert Einstein College of Medicine, Bronx, United States; [3]Menzies Institute for Medical Research University of Tasmania, Hobart, Australia

**Abstract** Mutations in the lysine demethylase 5 (KDM5) family of transcriptional regulators are associated with intellectual disability, yet little is known regarding their spatiotemporal requirements or neurodevelopmental contributions. Utilizing the mushroom body (MB), a major learning and memory center within the *Drosophila* brain, we demonstrate that KDM5 is required within ganglion mother cells and immature neurons for proper axogenesis. Moreover, the mechanism by which KDM5 functions in this context is independent of its canonical histone demethylase activity. Using in vivo transcriptional and binding analyses, we identify a network of genes directly regulated by KDM5 that are critical modulators of neurodevelopment. We find that KDM5 directly regulates the expression of *prospero*, a transcription factor that we demonstrate is essential for MB morphogenesis. Prospero functions downstream of KDM5 and binds to approximately half of KDM5-regulated genes. Together, our data provide evidence for a KDM5–Prospero transcriptional axis that is essential for proper MB development.

*For correspondence:
julie.secombe@einsteinmed.org

**Competing interests:** The authors declare that no competing interests exist.

## Introduction

Intellectual disability (ID) is reported to affect 1.5–3% of the global population and represents a class of neurodevelopmental disorders characterized by cognitive impairments that result in lifelong educational, social, and financial consequences for patients and their caregivers (*van Bokhoven, 2011*; *Leonard and Wen, 2002*). ID disorders are diagnosed during early childhood and are defined by an IQ score of less than 70 with deficits in adaptive behaviors (*Ropers, 2010*). However, despite these profound burdens, little is known regarding the pathogenesis of ID disorders, particularly how disruptions in genetic and neuronal regulatory programs contribute to cognitive and behavioral dysfunction.

Advances in comparative genomic hybridization and whole-exome sequencing have revealed strong associations between ID and mutations in genes encoding chromatin-modifying enzymes (*Elbert and Bérubé, 2014*; *Harmeyer et al., 2017*; *Kong et al., 2018*; *Parkel et al., 2013*). These proteins post-translationally modify chromatin by inserting or removing chemical moieties from histone tails to regulate DNA accessibility or alter the recruitment proteins needed to modulate transcriptional initiation or elongation (*van Bokhoven, 2011*; *Liefke et al., 2010*; *Liu et al., 2014*; *van Oevelen et al., 2008*; *Secombe et al., 2007*; *Vallianatos and Iwase, 2015*). One class of ID-associated chromatin modifiers is the lysine demethylase 5 (KDM5) family of transcriptional regulators, with mammals encoding four *KDM5* paralogs: *KDM5A*, *KDM5B*, *KDM5C*, and *KDM5D*. Loss-of-function mutations in *KDM5A*, *KDM5B*, and *KDM5C* are associated with ID, with genetic variants in *KDM5C* associated with a disorder known as mental retardation, X-linked, syndromic, Claes–Jensen type (MRXSCJ, OMIM# 300534).key.

The generation of *KDM5* knockout animal models has greatly assisted in our ability to investigate the neuromorphological and behavioral consequences of *KDM5* loss of function. Previous in vitro studies examining rat cerebellar granular neurons and pyramidal neurons of prepared mouse basolateral amygdala slices demonstrate that *Kdm5c* knockout results in dendritic spine abnormalities (*Iwase et al., 2016*). Similarly, loss-of-function mutations in *rbr-2*, the sole *Caenorhabditis elegans Kdm5c* ortholog, result in axonal growth and guidance defects (*Mariani et al., 2016*). Additionally, *Kdm5c* knockout mice display behavioral deficits that are analogous to those exhibited by patients with pathogenic *KDM5C* variants, such as increased aggression, learning and memory impairments, and decreased seizure thresholds (*Iwase et al., 2016*; *Scandaglia et al., 2017*). Together, these studies suggest that the neuromorphological and functional impairments resulting from loss of orthologous KDM5 proteins are likely to be attributed to altered gene expression within neurons.

KDM5 proteins demethylate trimethyl groups on lysine 4 of histone H3 (H3K4me3) via the enzymatic activity of their Jumonji C (JmjC) domains (*Liefke et al., 2010*; *Liu et al., 2014*; *Secombe et al., 2007*; *van Oevelen et al., 2008*). High levels of H3K4me3 near transcriptional start sites (TSS) are associated with actively transcribed genes, suggesting that KDM5 proteins can dynamically regulate transcription (*Greer and Shi, 2012*). Prevailing models linking alterations in KDM5 family protein function to ID suggest that loss of JmjC-mediated demethylase activity is a key driver of neuronal dysfunction (*Belalcazar et al., 2021*; *Mariani et al., 2016*; *Scandaglia et al., 2017*; *Vallianatos and Iwase, 2015*; *Vallianatos et al., 2020*; *Zamurrad et al., 2018*). For example, Vallianatos and colleagues have demonstrated that the neuronal and behavioral phenotypes observed in *Kdm5c* knockout mice can be rescued by reducing levels of the H3K4 methyltransferase KMT2A (*Vallianatos et al., 2020*). Similarly, *Drosophila* KDM5 acts in a demethylase-dependent manner to regulate long- and short-term olfactory memory (*Zamurrad et al., 2018*).

KDM5 family proteins can also regulate transcription independently of their demethylase activity by associating with other chromatin-modifying proteins (*Gajan et al., 2016*; *Lee et al., 2007*; *Lee et al., 2009*; *Nishibuchi et al., 2014*). For example, the HDAC complex member SIN3A has been shown in vitro to interact with *Drosophila* KDM5 and regulate overlapping subsets of genes. In mice, SIN3A is required for neuronal development (*Gajan et al., 2016*; *Witteveen et al., 2016*) with mutations in SIN3A associated with Witteveen–Kolk syndrome (OMIM# 613406), a neurodevelopmental disorder characterized by developmental delay and ID (*Witteveen et al., 2016*). Additionally, a subset of ID-associated *KDM5C* missense mutations have been shown in vitro not to affect H3K4me3 demethylase activity, yet alter transcriptional outputs (*Brookes et al., 2015*; *Vallianatos et al., 2018*). Collectively, these data provide strong evidence that disruption of KDM5 protein function may impact multiple transcriptional pathways critical to neuronal development and function.

Here, we utilize *Drosophila*, which encodes a single, highly conserved *KDM5* ortholog known as *kdm5* (previously known as *little imaginal discs* [*lid*]), to investigate the genetic and neuromorphological consequences of KDM5 loss during neurodevelopment. Our analyses focus on a group of neurons known as Kenyon cells, which form a bilateral, neuropil-rich structure known as the mushroom body (MB). The MB is essential for orchestrating a diverse repertoire of cognitive processes and is thus routinely used to study neuroanatomical changes associated with mutations in ID-related genes (*Androschuk et al., 2015*; *Aso et al., 2014*; *Dubnau et al., 2001*; *Heisenberg et al., 1985*). In fact, loss-of-function mutations in orthologous genes associated with ID, such as the Fragile X syndrome gene *fmr1*, the Down syndrome gene *dscam,* and the *ZC3H14* autosomal-recessive ID gene *dnab2*, result in severe morphological defects of the MB (*Hattori et al., 2007*; *Kelly et al., 2016*; *Michel et al., 2004*; *Zhan et al., 2004*).

The development of the MB is dependent on four MB neuroblasts (MBNBs) per hemisphere dividing asymmetrically throughout development. Each MBNB gives rise to another MBNB and a ganglion mother cell (GMC), which in turn divides symmetrically to form two Kenyon cells. Three subclasses of Kenyon cells give rise to the MB and are born in a highly regulated and sequential manner with tight temporal control. The first-born Kenyon cells are referred to as γ Kenyon cells and develop between the embryonic and mid-third-instar larval stage, giving rise to the γ lobes. The α′/β′ lobes are the next to develop, followed by the α/β lobes during the late larval and pupal stages. Notably, specification of these Kenyon cell subsets is transcriptionally regulated through the timed expression of a number of transcription factors (*Bates et al., 2010*; *Marchetti and Tavosanis, 2017*; *Syed et al., 2017*).

We show here that *kdm5* gene knockout and shRNA-mediated depletion of *kdm5* within GMCs and immature neurons both result in profound MB structural defects. Furthermore, using an in vivo transcriptomics-based approach, we identify subsets of genes within GMCs and immature Kenyon cells that are downregulated upon *kdm5* depletion. One such gene, *prospero* (*pros*), encodes a homeodomain-containing transcription factor that is required for axon pathfinding and growth in other neuronal cell types; however, its importance in regulating MB development remains unexplored. We find here that MB-specific knockdown of *pros* leads to aberrant MB formation and demonstrate that KDM5 directly binds to and regulates *pros* expression within GMCs and immature neurons. As approximately half of KDM5 regulated genes are bound by Pros, we propose a model by which KDM5 regulates the expression of Pros and its targets to promote proper MB formation. Our studies thus provide the first in vivo analysis of KDM5 within a specific cell population, revealing a key *kdm5-pros* genetic pathway critical for neurodevelopment.

## Results

### KDM5 is essential for proper MB morphology

To assess the neurodevelopmental consequences resulting from KDM5 loss, we examined MB morphology of animals that were homozygous for a *kdm5* null allele, *kdm5^140* (***Drelon et al., 2018***; ***Drelon et al., 2019***). As homozygous *kdm5^140* animals fail to eclose from their pupal cases, we performed our immunohistochemical analyses on pharate adults, which externally appear indistinguishable from wild-type animals (***Drelon et al., 2018***). Following a well-established classification scheme used by others (***Gombos et al., 2015***; ***Kelly et al., 2016***; ***Michel et al., 2004***), MB defects were categorized as impacting MB growth and/or guidance, with the former defined by a stunted, overextended, or absent lobe and the latter by a misprojected lobe. Staining using an antibody specific to the NCAM-like cell adhesion molecule fasciclin 2 (Fas2) revealed highly penetrant MB abnormalities, with ~70% of animals exhibiting growth or guidance defects of the α/β lobes (***Figure 1A, B***). Interestingly, the predominant phenotype observed in our analysis was overextension of the β lobes and/or stunting of the α lobes. Because animals specifically lacking KDM5 demethylase activity have phenotypically normal α/β lobes (***Zamurrad et al., 2018***), our data demonstrate that KDM5 is required for correct MB morphology, independent of its canonical enzymatic function.

To assist in our functional understanding of KDM5, we sought to investigate its expression throughout central nervous system (CNS) development. To facilitate these analyses, we used CRISPR/Cas9 to generate a strain containing a *3xHA*-tag fused to the endogenous locus of *kdm5* that results in wild-type levels of protein expression (***Figure 1C***). We also confirmed that KDM5:HA was expressed in adult MB Kenyon cells by co-expression of a nuclear membrane-localized GFP reporter, UNC-84:GFP, with the MB driver *OK107-Gal4* (***Figure 1D***). Immunostaining of the of wandering third-instar larvae revealed that KDM5 was localized to cortical nuclei while absent from neuropil-rich regions marked by the ubiquitous presynaptic active zone marker Bruchpilot (Brp) (***Figure 2A***). KDM5:HA localized to cortical nuclei across a variety of cell types, including neurons (***Figure 2B***), neuroblasts (NBs), and presumptive GMCs (***Figure 2C***). We also examined KDM5:HA expression in the adult brain, where KDM5:HA appeared to be similarly localized to cortical nuclei while absent from neuropil-rich regions, such as the antennal lobes and both the dorsolateral and ventrolateral protocerebra (***Figure 2D, E***). Given the broad expression pattern of KDM5:HA within a variety of cell types, KDM5 may regulate a range of neural processes, from NPC division and axonal growth to neuronal maturation and function.

### KDM5 is required within neural precursors and immature neurons for proper MB morphology

To define the functional requirements of KDM5 during MB development, we utilized an inducible *kdm5* shRNA transgene that we and others have shown effectively reduces KDM5 levels (***Chen et al., 2019***; ***Liu et al., 2014***; ***Navarro-Costa et al., 2016***). We first knocked down *kdm5* broadly within all MBNBs, MB-GMCs, and Kenyon cells throughout development using *OK107-Gal4* (***Figure 3A, B***). This resulted in significant neuromorphological defects of the α/β MB lobes, with the major phenotype being an overextension of the β lobes across the midline (***Figure 3C, D***). We next assessed the consequences of *kdm5* depletion within distinct and overlapping subsets of mature

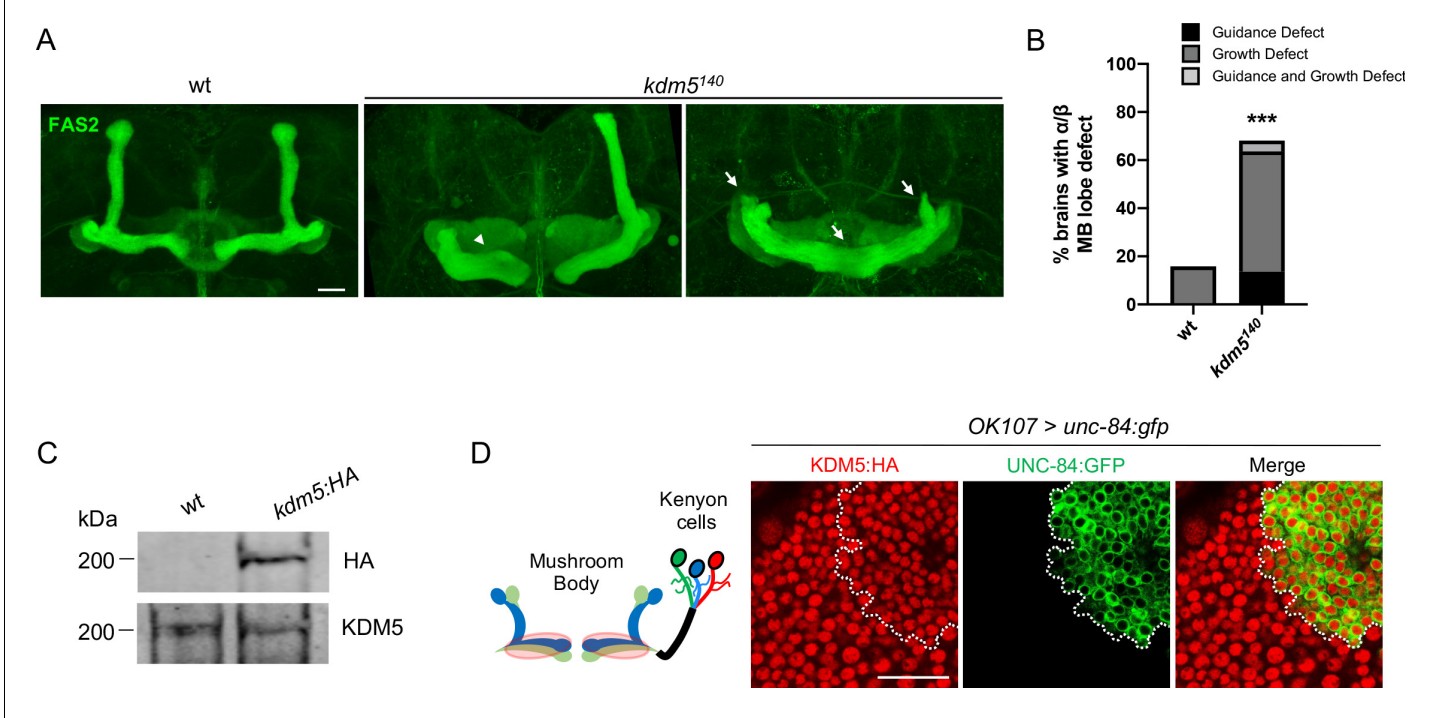

**Figure 1.** $kdm5^{140}$ pharate adults have neuromorphological defects of the mushroom body (MB). (A) Representative α/β lobe Z projections of pharate wild-type ($NP4707^{rev2}$ revertant) and $kdm5^{140}$ strains. Arrows indicate growth defects, and arrowheads indicate guidance defects. The α/β lobes are revealed with anti-fasciclin 2. (B) Quantification of α/β lobe defects in wild-type and $kdm5^{140}$ strains. $n$ = 19–22 (mean $n$ = 21). ***$p<0.001$ (chi-square test with Yates' correction). (C) Western blot of $w^{1118}$ and $kdm5:HA$ adult heads confirming wild-type expression levels of lysine demethylase 5 (KDM5): HA within our endogenously tagged $kdm5:HA$ strain. Anti-HA (top) and anti-KDM5 (bottom) loading control. (D) Schematic of an adult MB with its associated Kenyon cells (left). $OK107-Gal4$ is used to drive expression of UNC-84:GFP, an inner nuclear membrane GFP reporter, within MB neuroblasts, mushroom body-ganglion mother cells, and Kenyon cells (right). Brains are counterstained with anti-HA to demonstrate the presence of endogenously tagged KDM5:HA within Kenyon cell nuclei. Scale bars represent 20 μm.

Kenyon cells. Knocking down $kdm5$ using the mature Kenyon cell-specific Gal4 drivers $C708a$-, $c305a$-, $H24$-, and $201Y$-$Gal4$ reduced KDM5 levels, but failed to produce significant gross morphological defects of the α/β lobes (*Figure 3D*, *Figure 3—figure supplement 1*). KDM5 is therefore unlikely to be required exclusively within mature, post-mitotic Kenyon cells for proper α/β lobe development.

In contrast, knockdown of $kdm5$ using two independent NB-restricted *Gal4* driver lines, *worniu-Gal4* (*wor-Gal4*) and *inscuteable-Gal4* (*insc-Gal4*), resulted in profound α/β lobe defects (*Figure 3C, D*). Notably, a significant proportion of these brains simultaneously displayed both growth and guidance defects of the α/β lobes. Although *wor-Gal4* expression is NB specific, KDM5 depletion and GFP expression continued to be observed in presumptive GMCs and post-mitotic cells surrounding each NB (*Figure 3E*). This is likely attributed to perdurance of the Gal4 activator protein and/or the $kdm5$ shRNA. KDM5 could therefore be required within the NB, GMC, or even post-mitotically within the immature neuron for proper α/β Kenyon cell neurodevelopment.

To increase the resolution of our analysis, we leveraged the restricted expression pattern of *R71C09-Gal4*, which drives expression within GMCs and early-born neurons of the *Drosophila* larval CNS (*Figure 4A*; *Li et al., 2014*; *Marshall and Brand, 2017*; *Aughey et al., 2018*). *R71C09-Gal4* did not appear to drive expression within NBs, including MBNBs (*Figure 4A*). When $kdm5$ was knocked down using *R71C09-Gal4*, KDM5 was dramatically depleted within presumptive GMCs and immature neurons (*Figure 4B*). We additionally noted that the axons of newly born Kenyon cells, which traverse through the core fibers of the MB pedunculus, were also labeled by an mVenus reporter when driven by *R71C09-Gal4* (*Figure 4C*). Consistent with these observations, the labeled neurons were of α/β Kenyon cell origin, the last MB cell subtype to be born prior to eclosion (*Figure 4C*). Depletion of $kdm5$ using this driver resulted in profound α/β lobe defects, indicating that KDM5 is functionally

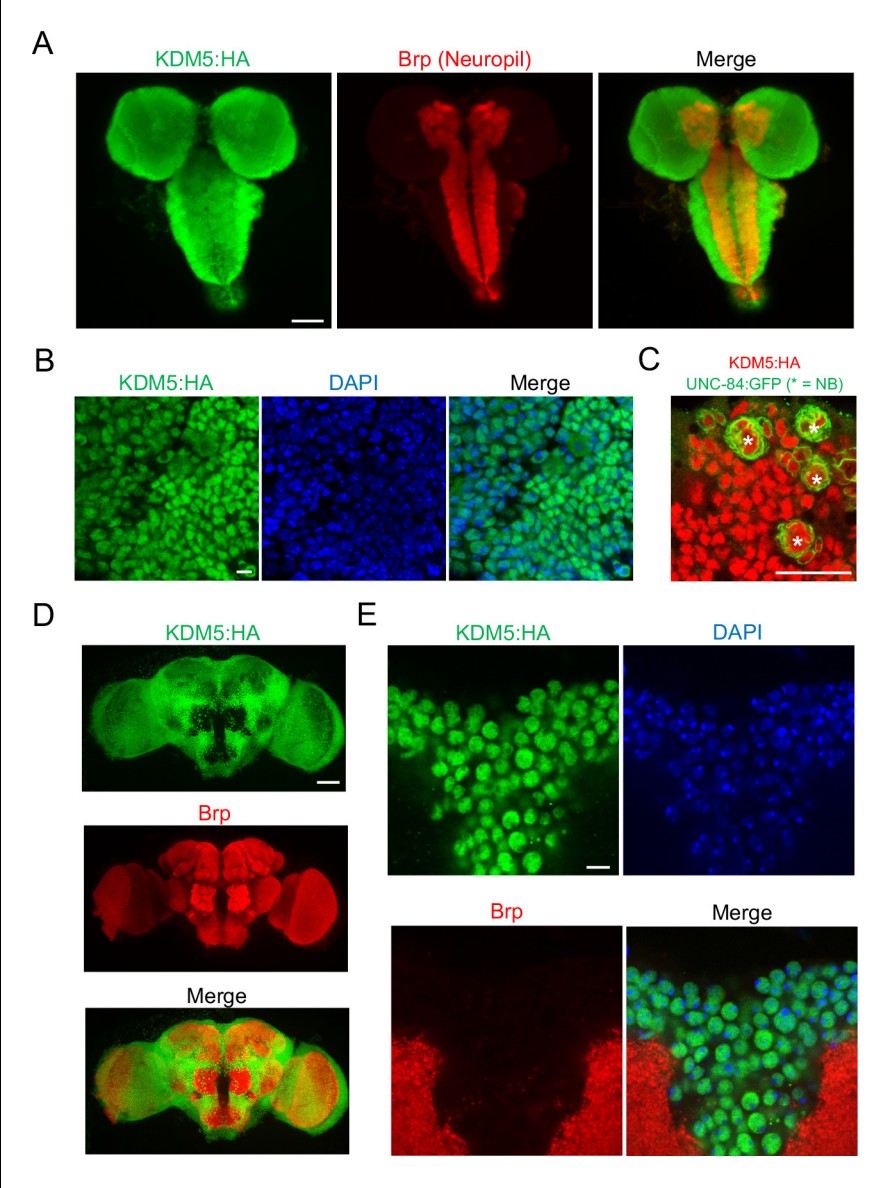

**Figure 2.** lysine demethylase 5 (KDM5) is broadly expressed in nuclei of the *Drosophila* larval central nervous system (CNS) and adult brain. (**A**) Maximal Z projection of third-instar larval CNS revealing broad expression of endogenously tagged KDM5:HA and stained with anti-HA and anti-Bruchpilot (anti-Brp). (**B**) Cortical region of a larval (WL3) brain lobe stained with anti-HA and DAPI, showing nuclear localization of endogenously tagged KDM5:HA. (**C**) Neuroblast (NB)-specific *wor-Gal4* driving expression of UNC-84:GFP to demonstrate endogenously tagged KDM5:HA expression within nuclei of WL3 central brain NBs (marked by \*) and presumptive ganglion mother cells. UNC-84:GFP perdures for approximately 2–3 cell divisions. (**D**) Maximal Z projection of an adult brain revealing broad expression of endogenously tagged KDM5:HA with anti-HA and counterstained for neuropil with anti-Brp. (**E**) Dorsoanterior cortical region of an adult brain with endogenously tagged KDM5:HA. The nuclear localization of KDM5:HA is revealed by anti-HA, DAPI, and anti-Brp staining. Scale bars represent 50 µm in (**A**) and (**D**), 5 µM in (**B**) and (**E**), and 20 µM in (**C**).

required within GMCs and immature MB neurons for proper axonal development (*Figure 4D*). To assess sufficiency, we used *R71C090-Gal4* to re-express *kdm5* in GMCs and immature neurons of *kdm5^{140}* animals and found a significant recue of the defects we had previously observed (*Figure 4E*).

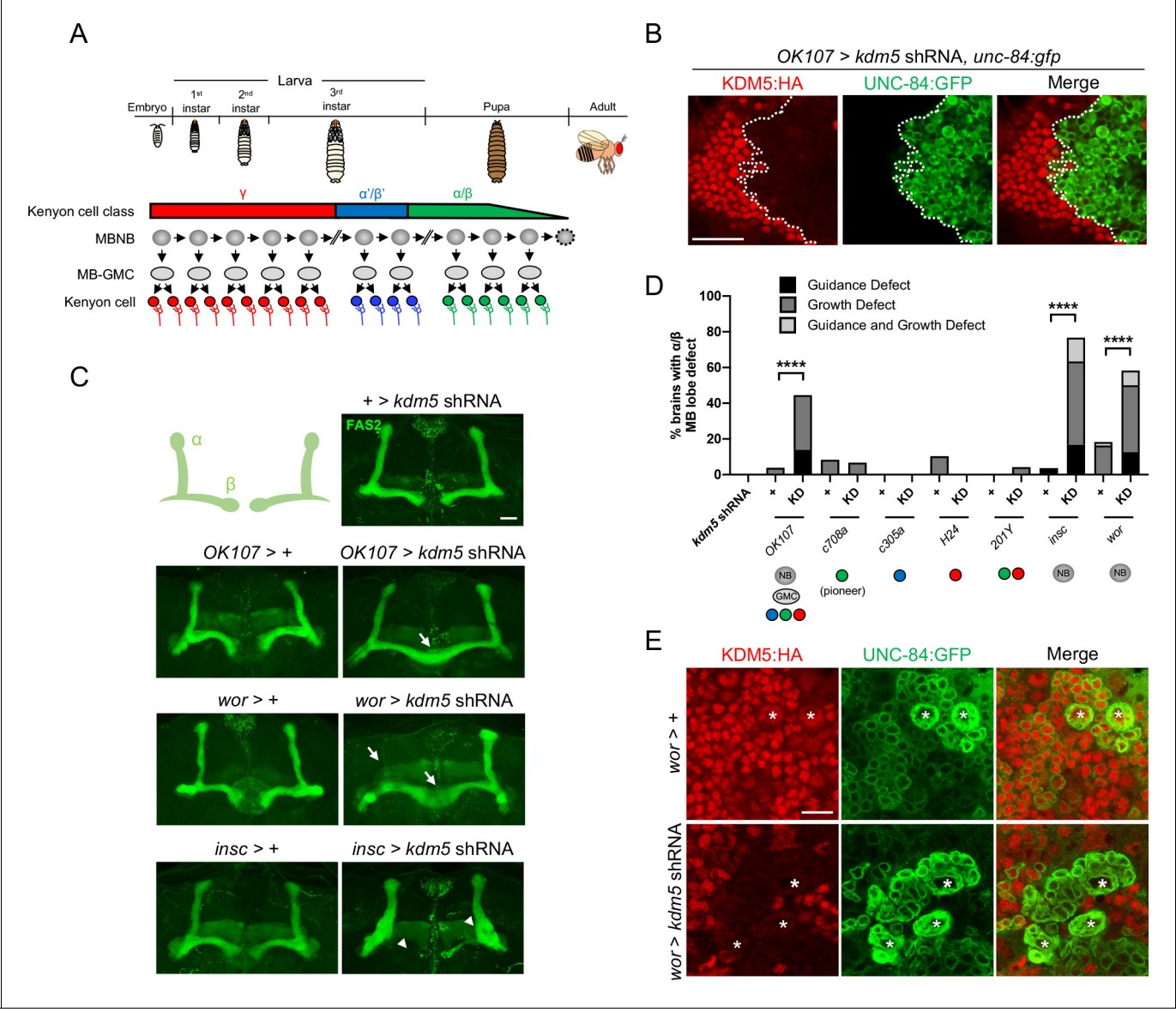

**Figure 3.** Depletion of lysine demethylase 5 (KDM5) within neural precursors results in neuroanatomical defects of the mushroom body (MB). (A) Schematic showing the sequential generation of distinct subclasses of Kenyon cells throughout development. MB neuroblasts (MBNBs) are eliminated via apoptosis (dotted line) immediately prior to eclosion. (B) *OK107-Gal4* driving expression of UNC-84:GFP to reveal shRNA-mediated KDM5:HA depletion within adult Kenyon cell nuclei. (C) Representative Z projections of adult *kdm5* knockdown animals exhibiting significant α/β lobe defects and their respective *kdm5* shRNA and *GAL4* controls. The antibody anti-fasciclin 2 is used to visualize α/β lobes. Arrows indicate growth defects, and arrowheads indicate guidance defects. (D) Quantification of α/β MB lobe defects in flies expressing *kdm5* shRNA driven by neural progenitor cell- and Kenyon cell-specific drivers. 'KD' indicates shRNA-mediated knockdown of *kdm5*. n = 16–49 (mean n = 29). ****p<0.0001 (chi-square test with Bonferroni correction). (E) Z projection of larval cortex revealing *wor-Gal4*-driven expression of *kdm5* shRNA and *unc-84:gfp transgenes*. KDM5:HA depletion is observed in presumptive ganglion mother cells and post-mitotic cells surrounding NBs (marked by asterisks). Scale bars represent 20 μm in (B) and (C) and 10 μm in (E).

The online version of this article includes the following figure supplement(s) for figure 3:

**Figure supplement 1.** Validation of *kdm5* knockdown within subpopulations of mature mushroom body (MB) neurons.

## Transcriptional profiling using targeted DamID (TaDa) analyses reveals KDM5-regulatory networks critical for neurodevelopment

To understand how KDM5 functions within GMCs and immature neurons to regulate MB development, we used TaDa to carry out in vivo transcriptional profiling in a cell-type-specific and

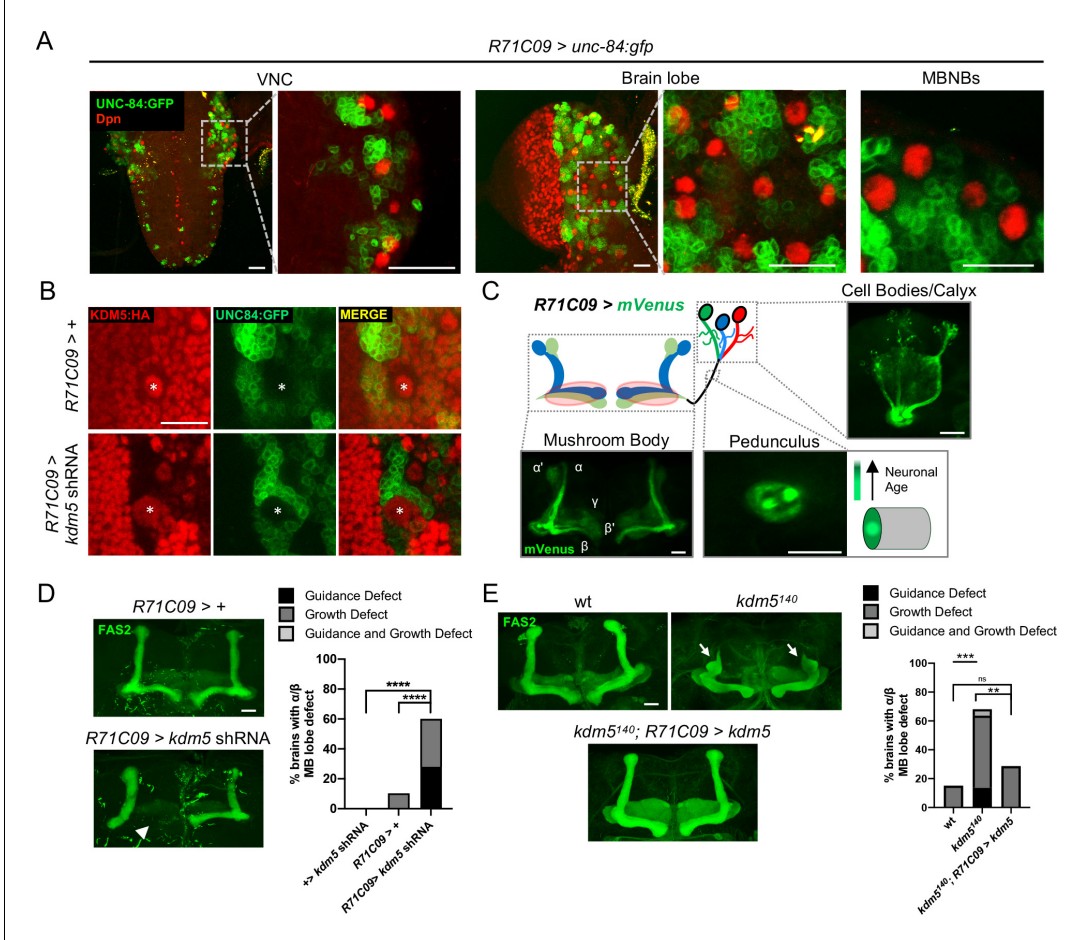

**Figure 4.** Expression pattern of *R71C09-Gal4* within ganglion mother cells and immature neurons of the *Drosophila* central nervous system. (**A**) Whole-mount Z projections of a larval ventral nerve cord (left), brain lobe (middle), and brain cortical region (right). Z projections show *R71C09-Gal4*-driven expression of UNC-84:GFP, counterstained with neuroblast (NB)-specific anti-Dpn. (**B**) Z projection of larval cortex revealing *R71C09-Gal4*-driven *expression of* UNC-84:GFP with and without *kdm5* shRNA. NBs are marked by an asterisk. (**C**) Optical sections of an adult mushroom body (MB) with its associated pedunculus, Kenyon cell bodies, and calyces expressing an *R71C09-Gal4*-driven mVenus reporter. *R71C09-Gal4* strongly drives mVenus expression in newly born neurons located within core fibers of the pedunculus. (**D**) Representative Z projections (left) and quantification (right) of adult α/β MB lobe defects in flies expressing *kdm5* shRNA driven by *R71C09-Gal4*. The antibody anti-fasciclin 2 (anti-Fas2) is used to visualize α/β lobes. n = 15–39 (mean n = 26). Arrowhead indicates a guidance defect. ****p<0.0001 (chi-square test with Bonferroni correction). (**E**) Representative α/β lobe Z projections and quantification of pharate wild-type (*NP4707^rev2* revertant), *kdm5^140*, and *kdm5^140; R71C09 > kdm5* rescue strains. Arrows indicate growth defects. The α/β lobes are revealed with anti-Fas2. n = 20–28 (mean n = 24). **p<0.01; ***p<0.001 (chi-square test with Bonferroni correction). Scale bars represent 20 μm.

temporally controlled manner (*Marshall et al., 2016a*; *Southall et al., 2013*). By expressing *Escherichia coli*-derived DNA adenine methyltransferase (Dam) fused to *Drosophila* RNA polymerase II (Pol II) in MB-GMCs and immature Kenyon cells via the Gal4/UAS system, we surveyed genomic regions with altered Pol II occupancy upon depletion of KDM5. Dam-Pol II methylates adenine residues (m6A) at GATC motifs in close proximity to Pol II-occupied DNA, providing a surrogate for actively transcribed loci when normalized to expression of Dam alone (*Doupé et al., 2018*; *Southall et al., 2013*). As such, we refer to genomic loci with altered Pol II occupancy as differentially expressed genes (DEGs).

We expressed Dam-Pol II or Dam in GMCs and immature neurons of late third-instar larvae and pupae of homozygous *kdm5^140* animals using the *R71C09-Gal4* driver (*Figure 5A*). Induction of the TaDa system during this developmental window predominantly resulted in the transcriptomic profiling of MB-GMCs and immature α/β Kenyon cells as these belong to one of the few cell lineages undergoing extensive neurogenesis during this time period and express *R71C09-Gal4* (*Technau and*

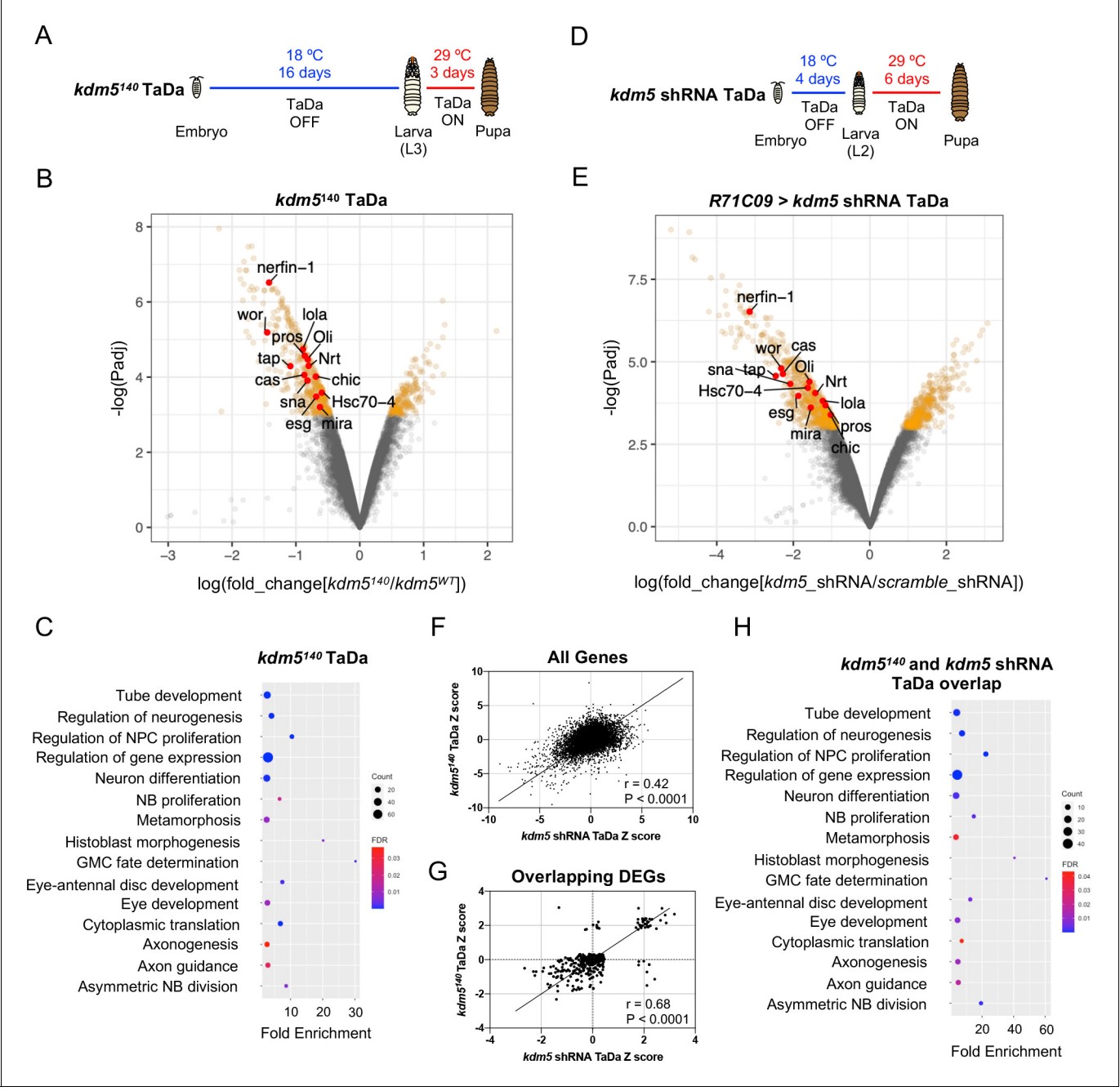

**Figure 5.** Transcriptome profiling of lysine demethylase 5 (KDM5)-depleted ganglion mother cells (GMCs) and immature neurons by targeted DamID (TaDa) reveals KDM5-regulatory networks critical for ganglion mother cell (GMC) proliferation and neurodevelopment. (**A**) Timeline of TaDa induction within GMCs and immature neurons of *kdm5*[140] WL3 animals and pupae. (**B**) Volcano plot of differentially expressed genes (DEGs) within GMCs and immature neurons of *kdm5*[140] animals compared to wild type. Genes with a false discovery rate (FDR) < 0.05 are in red, with those labeled involved in GMC proliferation and neurodevelopment. TaDa analyses were performed in quintuplicate. (**C**) Representative distribution of ontology terms for DEGs in *kdm5*[140] GMCs and immature neurons using a PANTHER Overrepresentation Test (Fisher's exact test with FDR < 0.05). (**D**) Timeline of TaDa and *kdm5* shRNA induction within GMCs and immature neurons of WL3 animals and pharate adults. (**E**) Volcano plot of DEGs within *kdm5* shRNA GMCs and immature neurons compared to those expressing a scrambled shRNA. Genes with an FDR < 0.05 are in red, with those labeled involved in GMC proliferation and neurodevelopment. TaDa analyses were performed in triplicate. (**F**) Correlation of Z scores between DEGs of *kdm5*[140] and *kdm5* shRNA TaDa datasets (Deming regression; p<0.0001). (**G**) Correlation of Z scores between overlapping DEGs of *kdm5*[140] and *kdm5* shRNA TaDa datasets (Deming regression; p<0.0001). (**H**) Representative distribution of ontology terms for DEGs in overlapping *kdm5*[140] and *kdm5* shRNA TaDa datasets utilizing a PANTHER Overrepresentation Test (Fisher's exact test with FDR < 0.05).

The online version of this article includes the following figure supplement(s) for figure 5:

*Figure 5 continued on next page*

Figure 5 continued

**Figure supplement 1.** kdm5<sup>140</sup> pharate adults expressing targeted DamID (TaDa) genetic elements present with significant mushroom body (MB) morphological defects.

**Figure supplement 2.** Knockdown of *myc* within Kenyon cells does not result in morphological defects of the mushroom body.

**Figure supplement 3.** Chromatin accessibility profiling using targeted DamID (CATaDa) analyses reveal minimal changes to chromatin accessibility within ganglion mother cells (GMCs) and immature neurons upon loss of *kdm5*.

**Figure supplement 4.** Adults expressing *R71C09-Gal4*-driven *kdm5* shRNA in tandem with targeted DamID (TaDa) genetic elements present with significant mushroom body morphological defects.

**Figure supplement 5.** Transcriptome analyses reveal that lysine demethylase 5 (KDM5)-regulated gene expression is tissue-specific.

*Heisenberg, 1982*; *Ito and Hotta, 1992*; *Truman and Bate, 1988*). Importantly, the temperature shifts required to induce the TaDa system in a kdm5<sup>140</sup> animals did not alter the frequency or type of structural MB defects observed (*Figure 5—figure supplement 1*).

Carrying out TaDa from five biological replicates, we identified a total of 636 DEGs using a statistical cutoff of FDR < 0.05, 438 of which were downregulated and 198 of which were upregulated (*Figure 5B*; *Supplementary file 1*). Consistent with previously published mRNA-seq data in *Drosophila* and other organisms, loss of KDM5 resulted in moderate changes to gene expression, with an average 2.4-fold decrease among downregulated genes and 2.1-fold increase among upregulated genes (*Drelon et al., 2018*; *Iwase et al., 2016*; *Liu et al., 2014*; *Liu and Secombe, 2015*; *Lloret-Llinares et al., 2012*; *Lopez-Bigas et al., 2008*; *Lussi et al., 2016*; *Zamurrad et al., 2018*). Thus, KDM5 likely functions within this cellular subpopulation to fine-tune gene expression across a number of pathways that are critical for MB development.

To determine if these dysregulated genes were enriched for functional categories, we used the program GeneOntology (*The Gene Ontology Consortium, 2019*; *The Gene Ontology Consortium, 2000*; *The Gene Ontology Consortium, 2019*), which utilizes the PANTHER Classification System (*Mi et al., 2013*) to mine gene ontology (GO) information. This revealed significant enrichment for categories involved in GMC fate determination (e.g., *pros, brat, mira, cas*), neural precursor cell proliferation (e.g., *pros, wor, ase, esg, sna, E(spl)mγ-HLH, E(spl)mβ-HLH*), axon guidance (e.g., *pros, dac, brat, Hr51, chic, tap, pdm3, nerfin-1, lola*), and cytoplasmic translation, among others (*Figure 5C*; *Supplementary file 2*). The finding that ribosomal protein genes, such as *RpS2, RpS24, RpS28b, RpL3, Rpl23, RpL39*, and *RpL41*, were altered in our kdm5<sup>140</sup> TaDa data is consistent with our previously reported RNA-seq data from adult heads of demethylase-dead *kdm5* strains showing that genes required for cytoplasmic translation were affected (*Zamurrad et al., 2018*). Interestingly, however, knockdown of the transcription factor *myc*, which regulates the expression of most ribosomal protein genes (*Bellosta and Gallant, 2010*; *Grewal et al., 2005*), in MBNBs, GMCs, and Kenyon cells using the *OK107-Gal4* driver, did not result in any gross morphological defects of the α/β lobes (*Figure 5—figure supplement 2*). These data show that our observed MB phenotypes are not due to the regulation of cytoplasmic translation by KDM5.

H3K4me3 marks have been shown to be associated with regions of accessible chromatin, which often contain regulatory DNA sequences such as promoters and enhancers (*Bhaumik et al., 2007*; *Li et al., 2012*; *Park et al., 2020*; *Thurman et al., 2012*). If KDM5 functions independent of its H3K4me3 demethylase activity within GMCs and immature neurons to regulate MB development, loss of KDM5 may have only minimal effects on chromatin accessibility within this cellular subpopulation. To measure changes to chromatin landscape upon KDM5 loss, we utilized chromatin accessibility profiling using targeted DamID (CATaDa) (*Aughey et al., 2018*; *Aughey et al., 2019*). Dam methylates regions of highly accessible chromatin, thus providing an in vivo surrogate for chromatin accessibility (*Aughey et al., 2018*; *Aughey et al., 2019*). Comparing Dam methylation levels in *R71C09-Gal4* expressing cells between kdm5<sup>140</sup> and wild-type animals revealed minor changes to chromatin accessibility at a cutoff of FDR < 0.01 (*Figure 5—figure supplement 3A, B*). Overlapping reduced accessibility regions with the downregulated genes from our TaDa analysis revealed only 17 genes associated with reduced chromatin accessibility in GMCs and immature neurons at a cutoff of FDR < 0.01 (*Figure 5—figure supplement 3C*). Although this represented a significantly enriched proportion of the total number of genes associated with changes in chromatin accessibility (p=4.787E-09), they were not enriched via GO analysis for any biological categories. Because animals lacking demethylase activity do not present with gross MB morphological defects (*Zamurrad et al.,*

*2018*), KDM5 functions to regulate MB morphology largely via demethylase-independent transcriptional mechanisms that do not dramatically alter chromatin accessibility.

Since *kdm5*$^{140}$ animals have chronic loss of *kdm5* in all tissues, the gene expression changes we observed may be the result of both cell-autonomous and non-cell-autonomous effects. To look directly at the KDM5-regulated transcriptome in GMCs and early-born neurons, we performed TaDa in animals expressing a *kdm5* shRNA transgene under the control of *R71C09-Gal4*. In addition to utilizing a Dam-only control, we also accounted for activation of the RNAi pathway by expressing a scramble shRNA transgene under identical conditions. To ensure that changes to gene expression were reflective of the severe MB phenotypes we had previously observed, we induced the expression of *kdm5* shRNA and *dam-pol II* for 6 days, beginning during the early second-instar larval (L2) stage, to allow for sufficient depletion of KDM5 within MB-GMCs and immature α/β Kenyon cells (*Figure 5D*). This temporally targeted knockdown strategy recapitulated the adult α/β lobe phenotypes we had observed for constitutive *kdm5* knockdown with *R71C09-Gal4* (*Figure 5—figure supplement 4*). From three biological replicates and a statistical cutoff of FDR < 0.05, we identified 1069 DEGs, 659 of which were downregulated and 410 of which were upregulated (*Figure 5E*; *Supplementary file 3*). Compared to the *kdm5*$^{140}$ TaDa, the greater number of DEGs from the *kdm5* RNAi TaDa could be attributed to the extended duration of the *kdm5* knockdown and induction of Dam-Pol II. Additionally, we observed larger changes to gene expression, with an average 5.5-fold decrease among downregulated genes and a 4.3-fold increase among upregulated genes (*Figure 5E*; *Supplementary file 3*). This could be ascribed to the acute loss of *kdm5* expression caused by RNAi-mediated depletion, which would decrease the likelihood of compensatory changes occurring.

To obtain a list of high-confidence genes regulated by KDM5, we compared the DEGs found in our *kdm5*$^{140}$ and *kdm5* knockdown TaDa datasets. This revealed a total of 335 overlapping dysregulated genes, 319 of which were up- or downregulated in both datasets (r = 0.68, p<0.0001) (*Figure 5F, G*; *Supplementary file 4*). GO analysis of the 319 similarly dysregulated genes revealed an even greater enrichment in categories related to GMC fate determination (*pros, mira, cas*), neural precursor cell proliferation (*wor, esg, pros, mira, sna*), and axon guidance (*pros, chic, lola, Oli, tap, Nrt, nerfin-1, Hsc70-4*), among others (*Figure 5H*; *Supplementary file 4*). Interestingly, only 17 of these genes overlapped with our previously published *kdm5*$^{140}$ wing disc RNA-seq data (*Drelon et al., 2018*; *Figure 5—figure supplement 5*). This nonsignificant overlap suggests that KDM5 functions in a predominantly tissue- and cell-specific manner to regulate the expression of downstream targets.

To identify biologically relevant functional networks for the 319 KDM5-regulated, high-confidence genes, we performed gene network and community clustering analyses (*Morris et al., 2011*; *Shannon et al., 2003*). These analyses revealed seven discrete networks with greater than two nodes (*Figure 6A*). Of these, two networks were highly enriched for genes implicated in MB development (*toy* and *tap*), neural precursor cell proliferation (*pros, ase, wor, mira, sna, esg, E(spl)mγ-HLH*, and *E(spl)mβ-HLH*), and axon growth and guidance (*pros, nerfin-1, tap, emc, Nrt*, and *elav*).

To determine which genes affected by loss of KDM5 were likely to be direct targets, we generated a transgenic *Drosophila* strain expressing Dam fused to KDM5 under the control of *UAS* (*UAS-dam-kdm5*). To match the developmental window utilized for our RNA pol II TaDa analyses, we expressed Dam-KDM5 or Dam in late third-instar larvae and pupae using the *R71C09-Gal4* driver. From six biological replicates, we identified a total of 4399 genes with KDM5 peaks using a statistical cutoff of FDR < 0.01 (*Supplementary file 5*). Of these, 120 genes overlapped with the 319 genes we previously identified as being KDM5-regulated and represented a significant enrichment (*Figure 6B, C*; p=6.396E-07). Of particular interest was one gene, *prospero* (*pros*), which encodes a homeodomain-domain-containing transcription factor that is a critical regulator of axon pathfinding and growth (*Doe et al., 1991*; *Froldi et al., 2015*; *Tea et al., 2010*; *Vaessin et al., 1991*). Our Dam-KDM5 occupancy data showed that KDM5 binds to TSS within the *pros* locus (*Figure 7A*) and, together with our Pol II TaDa data, suggest that KDM5 directly regulates *pros* expression.

## The transcription factor *pros* genetically interacts with *kdm5* and is required for proper MB development

If KDM5 functions upstream of Pros to regulate its expression, it is likely that KDM5 may also indirectly regulate the expression of Pros targets. To explore this possibility, we compared our analyses

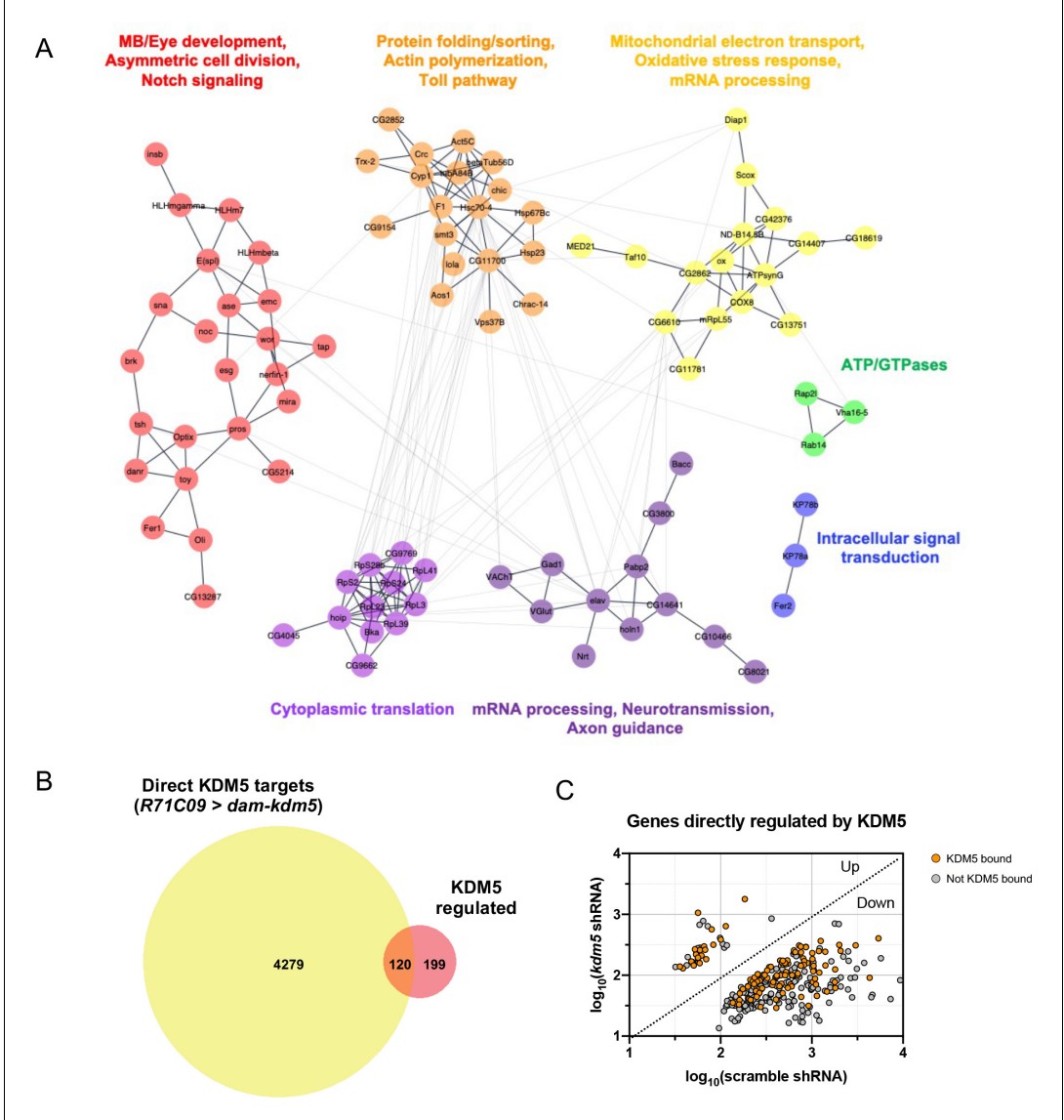

**Figure 6.** Network of known and predicted interactions between lysine demethylase 5 (KDM5)-regulated aenes and analysis of direct KDM5 targets. (**A**) Gene network analysis and community clustering were performed using Cytoscape with a minimum confidence score of 0.4. Networks with greater than two nodes are shown and color-coded based on cluster. Labels indicate general categories of overlapping *kdm5*[140] and *kdm5* shRNA targeted DamID (TaDa) differentially expressed genes (DEGs) within each cluster. (**B**) Venn diagram illustrating intersection of similarly dysregulated *kdm5*[140] and *kdm5* shRNA overlapping DEGs with direct KDM5 targets from *R71C09 > dam-kdm5* TaDa. (**C**) Analysis of the 319 similarly dysregulated *kdm5*[140] and *kdm5* shRNA overlapping DEGs (with values plotted from the *kdm5* shRNA TaDa). Direct KDM5 targets are labeled in orange. Venn diagram created with BioVenn (***Hulsen et al., 2008***).

with published Dam-Pros occupancy data within GMCs and immature neurons (***Liu et al., 2020***; Gene Expression Omnibus [GEO] accession number GSE136413). Strikingly, 45% (142 of 319) of KDM5-regulated genes were bound by Pros (p=2.20E-16, Fisher's exact test) (***Figure 7B, C***; ***Supplementary file 6***). In addition, a majority of Pros-bound targets were not co-bound by KDM5 (***Figure 7B***). These data suggest that Pros could be a key mediator of KDM5 function in GMCs and immature neurons. Our data also implicate Pros in the activation of genes as the majority of Pros-bound genes were downregulated upon KDM5 depletion (***Figure 7C***).

While roles for Pros in regulating NB and GMC asymmetric division are well-described (***Choksi et al., 2006***; ***Doe et al., 1991***; ***Vaessin et al., 1991***), its role in MB development remains unexplored. *R71C09-Gal4*-mediated knockdown of *pros* resulted in early larval lethality, preventing

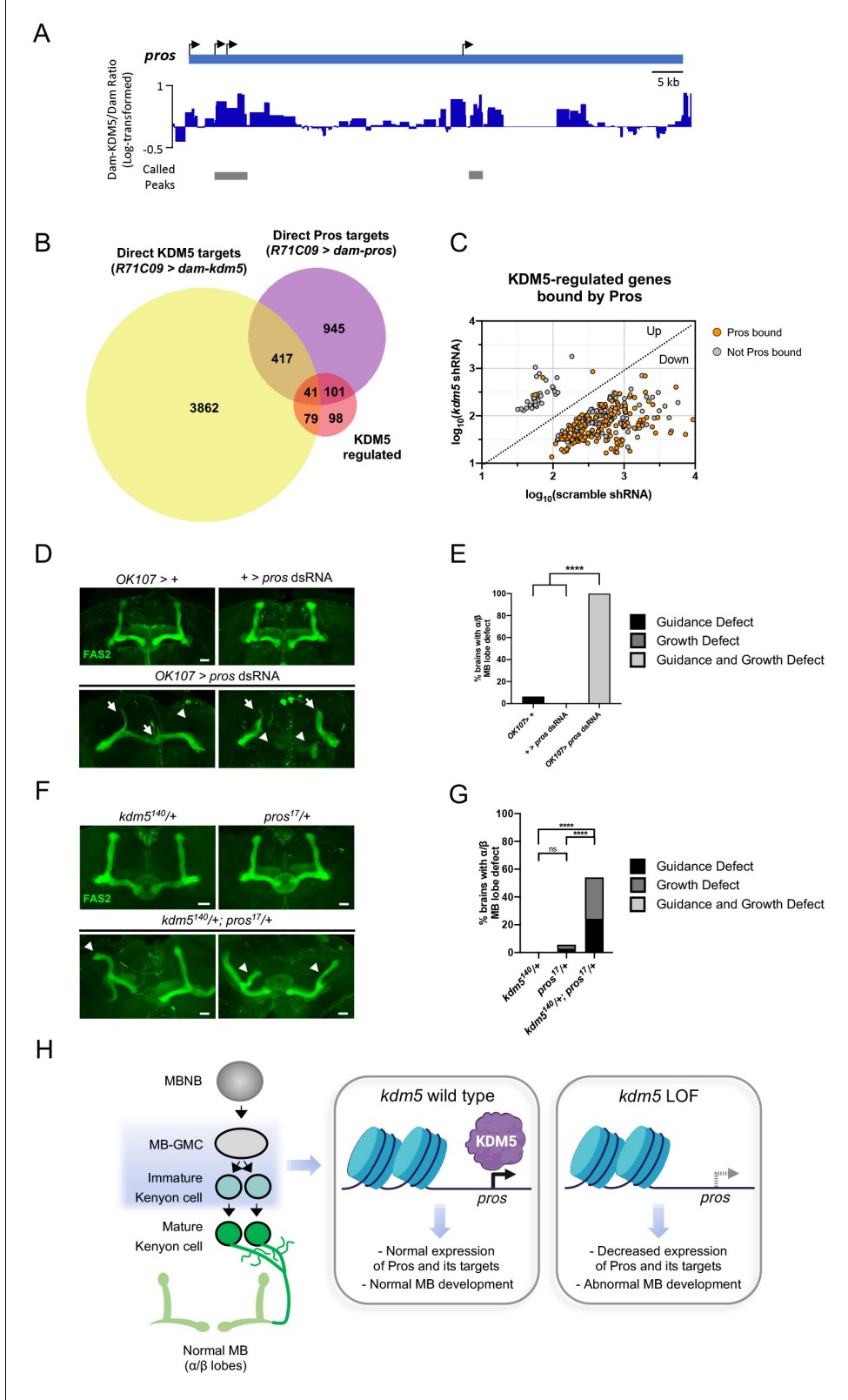

**Figure 7.** Neuromorphological and transcriptomic analyses reveal a genetic interaction between prospero and lysine demethylase 5 (KDM5). (**A**) Integrative genomics viewer (IGV) plot showing average KDM5 occupancy at *pros* transcriptional start sites across six replicates. Scale bars represent the log2 ratio change between Dam-KDM5 and Dam samples. Called peaks are indicated by gray bars. (**B**) Venn diagram illustrating intersection of

*Figure 7 continued on next page*

*Figure 7 continued*

similarly dysregulated *kdm5*[140] and *kdm5* shRNA overlapping differentially expressed genes (DEGs) with Dam-KDM5 direct binding data and a previously published Dam-Pros targeted DamID (TaDa) binding dataset (Fisher's exact test, p=2.20E-16 for KDM5-dysregulated and Pros gene overlap) (*Liu et al., 2020*). (C) Analysis of the 319 similarly dysregulated *kdm5*[140] and *kdm5* shRNA overlapping DEGs (with values plotted from the *kdm5* shRNA TaDa). Direct Pros targets from the previously published Dam-Pros TaDa dataset (*Liu et al., 2020*) are labeled in orange. (D) Representative Z projections of *OK107 > pros* RNAi adults exhibiting significant α/β lobe defects. The α/β lobes are revealed by anti-fasciclin 2 (Fas2). (E) Quantification of α/β mushroom body (MB) lobe defects in flies expressing *pros* shRNA driven by *OK107-Gal4*. n = 16–19 (mean n = 17). ****p<0.0001 (chi-square test with Bonferroni correction). (F) Representative Z projections of representative *kdm5*[140]/+ and *pros*[17]/+ heterozygous adult α/β MB lobes (top) and *kdm5*[140]/+; *pros*[17]/+ transheterozygous adult α/β MB lobes (bottom). The α/β lobes are revealed by anti-Fas2. (G) Quantification of α/β MB lobe defects in *kdm5*[140]/+; *pros*[17]/+ transheterozygous adults and heterozygous controls. n = 28–37 (mean n = 34). ****p<0.0001 (chi-square test with Bonferroni correction). (H) Model proposing a genetic interaction between *pros* and *kdm5* within ganglion mother cells (GMCs) and immature α/β Kenyon cells. KDM5 binds to the *pros* locus and positively regulates its transcription. Loss of *kdm5* leads to downregulation of *pros* and its targets, resulting in defects to MB neurodevelopment and cognitive function. Image was created with BioRender and Venn diagrams with BioVenn (*Hulsen et al., 2008*). Scale bars represent 20 µm.

us from examining MB phenotypes. However, depleting Pros within the smaller cellular population of MB progenitor and Kenyon cells using *OK107-Gal4* allowed for the formation of viable adults. Analyses of these adult brains revealed a complete disruption of MB morphology, with both growth and guidance defects present in all animals observed (*Figure 7D, E*).

As *kdm5* or *pros* depletion within MB progenitor and Kenyon cells each independently resulted in significant axonal defects, we next assessed whether *kdm5* and *pros* functioned within the same genetic pathway. To test this, we generated animals heterozygous for null alleles of both *kdm5* and *pros*, thereby simultaneously reducing levels of both proteins. Animals heterozygous for either *kdm5*[140] or *pros*[17] did not display significant MB defects. In contrast, adults that were transheterozygous for *kdm5*[140] and *pros*[17] displayed MB abnormalities (*Figure 7F, G*). These data are consistent with KDM5 and Pros functioning synergistically to affect MB development. KDM5 is therefore likely to function within GMCs and immature neurons to modulate levels of *pros* and its downstream targets, thereby regulating MB development (*Figure 7H*).

## Discussion

Here, we utilize in vivo transcriptional and binding approaches coupled with neuromorphological analyses to demonstrate that KDM5 is essential for the development of the adult MB. Although previous work in murine models has shown that knockout of the *kdm5* ortholog *Kdm5c* leads to dendritic spine abnormalities in cortical brain regions, these studies utilized ubiquitous knockout systems that were not amenable to spatiotemporal genetic manipulation (*Iwase et al., 2016*; *Vallianatos et al., 2020*). *Drosophila*, however, provides a genetically tractable system to modulate gene expression in a spatiotemporally restrictive manner. Using the MB as a model for neuronal development, we demonstrate that KDM5 expression within GMCs and immature neurons is necessary and sufficient for proper neuropil development. Depletion of KDM5 within mature subpopulations of Kenyon cells did not affect gross MB structure, suggesting that there exists a critical neurodevelopmental window during which KDM5 is required to regulate genetic programs essential for axonal growth and guidance. Notably, the MB growth and guidance defects we observed upon depletion of KDM5 phenocopied those seen in other *Drosophila* ID models, such as the *fmr1* and *dnab2* RNA binding protein mutants (*Kelly et al., 2016*; *Michel et al., 2004*). It is thus possible that ID genes may function synergistically within common neurodevelopmental pathways to affect neuronal architecture and function.

We additionally performed RNA pol II TaDa analyses within GMCs and immature neurons to survey gene expression changes resulting from loss of *kdm5*. Although previous transcriptomic studies have utilized extracted RNA from dissected tissues or from neuronal or fibroblast cultures as input for RNA-seq (*Brookes et al., 2015*; *Iwase et al., 2016*; *Liu and Secombe, 2015*; *Scandaglia et al., 2017*; *Vallianatos et al., 2020*; *Wei et al., 2016*; *Zamurrad et al., 2018*), this is the first study

describing changes to gene expression resulting from cell-specific loss of *kdm5*. This technique allowed us to survey gene expression changes within GMCs and immature neurons of *kdm5* null and cell-type-specific knockdown animals, revealing a number of DEGs associated with neurodevelopment. It is important to note, however, that since we induced expression of *UAS-dam* and *UAS-dam-Pol II* beginning during the late larval stages, a subset of our data may reflect changes to gene expression within GMCs and immature neurons that are not part of the MB lineage. Nevertheless, TaDa induction during the larval stage was brief (<24 hr at 29°C), with the vast majority of transcriptomic changes occurring within MB-GMCs and immature Kenyon cells, which are some of the few cell types able to proliferate up to 96 hr after pupal formation (*Truman and Bate, 1988*).

Our data suggest that one means by which KDM5 functions in neurons is by binding to and regulating the expression of the homeodomain-containing transcription factor Pros. Pros is a cell fate determinant that is expressed in most neuronal precursors and immature neurons and is involved in cell cycle exit, neuronal differentiation, and axonal development (*Choksi et al., 2006*; *Doe et al., 1991*; *Froldi et al., 2015*; *Tea et al., 2010*; *Vaessin et al., 1991*). Loss of *pros* results in axonal routing defects within embryonic motor and sensory neurons (*Vaessin et al., 1991*) and can lead to the miswiring of olfactory projection neuron dendrites (*Miyoshi et al., 2015*; *Tea et al., 2010*). However, despite its demonstrated importance in other neuronal contexts, a role for *Drosophila* Pros or its orthologs in regulating axonal growth and guidance programs has remained largely uncharacterized.

Pros is evolutionarily well-conserved, with the mammalian ortholog PROX1 having been shown to promote neuronal differentiation and migration. Specifically, studies utilizing chick and mouse neural progenitor cells demonstrate that PROX1 is essential for cell cycle exit and differentiation of neuronal precursors via the transcriptional repression of *Notch1* (*Kaltezioti et al., 2010*; *Lavado et al., 2010*). Additionally, studies leveraging conditional knockout mouse strains have shown that PROX1 regulates GABAergic cortical interneuron migration during embryonic and postnatal development (*Miyoshi et al., 2015*). However, despite its demonstrated importance in other neuronal contexts, a role for PROX1 or its orthologs in regulating axonal growth and guidance programs has remained largely uncharacterized. Here, we demonstrate that depletion of *Drosophila* Pros within MBNBs, MB-GMCs, and immature Kenyon cells leads to severe MB growth and guidance defects. Importantly, Pros is expressed primarily in neural precursors and immature neurons at a time when KDM5 function is critical for MB development (*Doe et al., 1991*; *Vaessin et al., 1991*).

Our transcriptomic data further reveal that a number of Pros interactors are dysregulated upon *kdm5* depletion. For example, the direct Pros target *nerfin-1*, which encodes a zinc finger transcription factor required for early axon pathfinding by most CNS neurons in the *Drosophila* embryo (*Kuzin et al., 2005*; *Liu et al., 2020*), was significantly downregulated. Additionally, genes encoding Toy and Tap, which have been shown to regulate MB morphology (*Furukubo-Tokunaga et al., 2009*; *Yuan et al., 2016*), were similarly affected. Transcriptional dysregulation of *pros* and its targets may thus provide important molecular insight into the neuronal phenotypes associated with *kdm5* loss of function.

Although we demonstrate that KDM5 binds to the *pros* gene and is needed for its activation, we do not know the precise mechanism by which this occurs. We predict, however, that it is independent of its canonical histone demethylase activity as animals lacking this enzymatic function have MBs that are phenotypically indistinguishable from those of wild type (*Zamurrad et al., 2018*). Because flies lacking KDM5 histone demethylase activity have cognitive deficits (*Zamurrad et al., 2018*), it is likely that KDM5 regulates neuronal development and function via multiple distinct mechanisms. Consistent with this model, several ID-associated mutations in KDM5C have been shown to alter H3K4me3-directed enzymatic activity in vitro, whereas others do not (*Brookes et al., 2015*; *Vallianatos et al., 2018*). Based on these data and the results reported here, it is likely that ID-causing mutations in human *KDM5A*, *KDM5B*, or *KDM5C* might fall into three classes: the first may only affect KDM5 demethylase activity, the second may only affect non-enzymatic activities necessary for transcriptional regulation, and the third may affect both. The extent to which canonical and non-canonical activities are disrupted may, indeed, correlate with the severity of the cognitive deficit or presence of syndromic features. Future work will further elucidate how individual patient mutations may disrupt these transcriptional regulatory mechanisms to influence neuronal and behavioral outputs, thus providing a promising strategy for predicting and treating patients with pathogenic variants in *KDM5* family genes.

## Materials and methods

### Resource availability

#### Lead contact

Further information and requests for resources and reagents should be directed to and will be fulfilled by the Lead Contact, Julie Secombe (Julie.Secombe@einsteinmed.org).

#### Materials availability

The *kdm5:HA* and *UAS-LT3-dam-kdm5* strains generated in this study are available from the Lead Contact without restriction. DamID-Seq data have been deposited in the GEO under accession number GSE156010 for Dam-Pol II TaDa and GSE166116 for Dam-KDM5 TaDa.

#### Code availability

The code supporting the current study is available at https://github.com/owenjm/polii.gene.call (*Marshall et al., 2013*), https://github.com/owenjm/damidseq_pipeline (*Marshall and Brand, 2015b*), and https://github.com/owenjm/find_peaks (*Marshall et al., 2016b*).

### Fly strains and genetics

A detailed list of the genotypes of the flies used in each figure is included in the Key Resources Table in the Appendix.

All *GAL4* and *GMR GAL4* lines were generated at the Janelia Research Campus/ HHMI (*Pfeiffer et al., 2008*; *Jenett et al., 2012*) and were obtained from the Bloomington *Drosophila* Stock Center (BDSC) at Indiana University.

The following transgenes were used: *UAS-kdm5-shRNA* (RRID:BDSC_35706), *20XUAS-IVS-CsChrimson.mVenus* (RRID:BDSC_55136), *UAS-tub-GAL80$^{ts}$* (RRID:BDSC_7108), *UAS-pros-dsRNA* (RRID:BDSC_42538), and *UAS-myc-dsRNA* (VDRC stock# KK106066). The *UAS-LT3-dam* and *UAS-LT3-dam-pol II* lines were kindly shared by Andrea Brand (U. Cambridge, Gurdon). The *5XUAS-unc84-2Xgfp* line was kindly shared by Gilbert Henry and Todd Laverty (HHMI Janelia). The *kdm5$^{140}$* mutant allele and *UAS-kdm5* transgene have been previously described (*Drelon et al., 2018*; *Secombe et al., 2007*).

### Cloning and transgenesis

To tag the endogenous *kdm5* locus with three in-frame *HA* epitope tags, we used CRISPR/Cas9-mediated knock-in. The *HA* epitopes and the homology arms, carrying a synonymous mutation for the PAM sequence, were PCR amplified from a clone containing the *kdm5* locus from the wild-type strain *w$^{1118}$* (pattB.gkdm5; *Navarro-Costa et al., 2016*). The donor DNA repair template consisted of three PCR fragments cloned, by In-Fusion HD (Takara, Bio), into the pHD-ScarlessDsRed vector (Addgene plasmid #51434). AarI and SapI enzymes were used to linearize the plasmid. The fly-CRISPR Optimal Target Finder tool was used to select the target genomic cleavage and design the gRNA (http://targetfinder.flycrispr.neuro.brown.edu). The top and bottom oligos for the gRNA were phosphorylated and annealed using T4 polynucleotide kinase (NEB) and then cloned into the pU6 vector (Addgene plasmid #53062), which was linearized with BpiI (NEB). The gRNA and donor DNA were sent to the BestGene for injection into embryos expressing Cas9 in the germline (*y$^1$, M{vas-Cas9.RFP-}ZH-2A, w$^{1118}$*; RRID:BDSC_55821). To remove the DSRed cassette, transformed flies were balanced and then crossed with flies carrying piggyBac transposase (RRID:BDSC_32073). Flies lacking expression of RFP were recovered and homozygous stocks were sequenced for the *kdm5* locus. The correct removal of the DSRed cassette and presence of the *3xHA* tag were confirmed by PCR sequencing and western blot.

To generate *UAS-LT3-dam-kdm5* flies, *kdm5* cDNA was PCR amplified from pB-Lid and cloned into the pUAST-attB-LT3-NDam vector (*Marshall et al., 2016a*). pUAST-attB-LT3-NDam was linearized with NotI and XbaI, and the cut plasmid and PCR insert were assembled by In-Fusion HD (Takara, Bio). The construct was sequenced and sent to BestGene for injection into embryos carrying the attP2 landing site (RRID:BDSC_8622).

## Immunohistochemistry

For fixation of late third-instar larval brain tissue, the CNS was first removed via dissection in cold phosphate buffered saline (PBS, pH 7.4) and allowed to incubate in fixation buffer (4% paraformaldehyde in PBS) at room temperature (RT) for 40 min. After washing for three cycles of 15 min in PBS with 0.2% Triton X-100 (0.2% PBT), brains were incubated with blocking buffer (5% normal goat serum in 0.2% PBT) for 30 min at RT. The brains were then incubated with primary antibodies for 2 days at 4°C and washed for three cycles of 15 min in 0.2% PBT. Secondary antibodies were then added and brains were incubated for 2 days at 4°C. After being washed in PBS for three cycles of 15 min at RT, brains were incubated in a drop of Vectashield mounting medium (Vector Laboratories, H-1000) or DAPI Fluoromount G (SouthernBiotech, OB010020) overnight at 4°C. Brains were then mounted on glass slides (Superfrost Plus, Fisherbrand), flanked by glass spacers and covered with a final glass coverslip before being used for image analysis.

A similar protocol was followed for immunostaining of pharate adult and 3- to 5-day-old adult brain tissue with the following exceptions. Pharate adult heads or whole adult animals were allowed to incubate in fixation buffer (4% paraformaldehyde in 0.2% PBT) at 4°C for 3 hr. Heads or whole animals were then washed in 0.2% PBT for three cycles of 15 min at RT. Brains were then dissected from the fixed tissue and blocked for 30 min at RT. Antibody incubation and mounting was identical to that described above.

The following primary antibodies were used: mouse anti-brp (1:50, DSHB cat# nc82, RRID:AB_2314866), mouse anti-Fas2 (1:25, DSHB cat# 1D4 anti-fasciclin II, RRID:AB_528235), rabbit anti-HA (1:100, Cell Signaling Technology cat# 3724; RRID:AB_1549585), mouse anti-HA (1:100, Cell Signaling Technology cat# 2367, RRID:AB_10691311), and rat anti-Dpn (1:100, Abcam cat# ab195173; RRID:AB_2687586). Primary antibodies were prepared in 5% NDS/0.2% PBT with 0.02% NaN$_3$. The following secondary antibodies were used: goat anti-mouse Alexa-488 (1:500, Thermo Fisher Scientific cat# A32723; RRID:AB_2633275), goat anti-mouse Alexa-568 (1:500, Thermo Fisher Scientific cat# A11004; RRID:AB_2534072), goat anti-rabbit Alexa-488 (1:500, Thermo Fisher Scientific cat# A11034; RRID:AB_2576217), goat anti-rabbit Alexa-568 (1:500, Thermo Fisher Scientific cat# A11004, RRID:AB_2534072), and goat anti-rat Alexa 568 (1:500, Thermo Fisher Scientific cat# A11077; RRID:AB_2534121). All secondary antibodies were diluted in 5% NDS/0.2% PBT.

## Western blotting

Western analysis was carried out as previously described (*Drelon et al., 2019*). Briefly, 3- to 5-day-old adult fly heads were homogenized in 2× NuPAGE LDS sample buffer, sonicated for 10 min, treated with DTT, run on a 4–12% Bis-Tris 1 mm gel, and transferred to a PVDF membrane. The following primary antibodies were used: rabbit anti-KDM5 (1:1000, Secombe J; Genes Dev. 2007; cat# lid, RRID:AB_2569502) and mouse anti-HA (1:1000, Cell Signaling Technology cat# 2367, RRID:AB_10691311). The following secondary antibodies were used: IRDye 680RD donkey anti-mouse IgG (1:8000; LI-COR Biosciences cat# 925-68072, RRID:AB_2814912) and IRDye 800CW donkey anti-rabbit IgG (1:8000; LI-COR Biosciences cat# 926-32213, RRID:AB_621848). Blots were scanned and processed using a LI-COR Odyssey Infrared scanner.

## Image acquisition

All tissue images were taken on a Leica SP8 confocal microscope using either a ×20 air lens (N.A. = 0.75 air, W.D. = 0.64 mm) or a ×63 immersion lens (N.A. = 1.4 oil, W.D. = 0.14 mm). All MB confocal stacks were taken under either ×2 or ×2.5 zoom, in a 1024 × 1024 configuration and using 1 µm resolution. Image stacks were processed with Figi (ImageJ). Figures were composed using Microsoft Powerpoint.

## Targeted DamID and analyses

To profile Pol II occupancy in GMCs and immature neurons of *kdm5*$^{140}$ pupae, *kdm5*$^{140}$/*CyO:gfp; UAS-LT3-dam* or *kdm5*$^{140}$/*CyO:gfp; UAS-LT3-dam-pol II* flies were crossed with *kdm5*$^{140}$/*CyO:gfp; R71C09-Gal4* flies and were allowed to lay eggs overnight for 12–14 hr at 25°C. Embryos were then moved to 18°C for a period of 16 days, and GFP-negative wandering third-instar larvae were transferred to a restrictive temperature of 29°C for 3 days to induce the expression of the *UAS-dam* and *UAS-dam-pol II* transgenes. As a parallel control, *kdm5* wild-type animals carrying *UAS-LT3-dam* or

*UAS-LT3-dam-pol II* were crossed with *R71C09-Gal4* and were also allowed to lay eggs for 24 hr at 25°C. Embryos were then moved to 18°C for 9 days and subsequently transferred to 29°C for 3 days.

For *kdm5* knockdown experiments, flies carrying either *UAS-LT3-dam; UAS-kdm5_shRNA* or *UAS-LT3-dam-pol II; UAS-kdm5_shRNA* were crossed with flies carrying *UAS-tubulin-Gal80^ts; R71C09-Gal4*. Parallel crosses were also performed with flies bearing a scrambled shRNA sequence in place of *kdm5* shRNA. All flies were allowed to lay eggs overnight for 12–14 hr at 25°C. Embryos were then transferred to 18°C for 4 days and then subsequently transferred to 29°C for 6 days.

To determine genomic regions directly bound by KDM5 within GMC and immature neurons, *UAS-LT3-Dam* or *UAS-LT3-Dam-kdm5* animals were crossed with flies carrying *tubulin-Gal80^ts; R71C09-Gal4* and allowed to lay eggs for 12–14 hr at 25°C. Embryos were transferred to 18°C for 9 days and subsequently moved to 29°C for 3 days.

Tissue processing for all TaDa experiments was performed as previously described in *Marshall et al., 2016a* with the following modifications. A total of 40 pupae of each genotype were homogenized in 500 mM EDTA followed by DNA extraction using the Zymo Quick-DNA Miniprep Plus Kit. DpnI digestion, PCR adaptor ligation, DpnII digestion, and PCR amplification were performed as described. DNA was sonicated using a Diagenode Bioruptor for 8–10 cycles (5 min at high power, 30 s on/30 s off) and analyzed using an Agilent Bioanalyzer. DamID adaptor removal and DNA cleanup were performed as previously described (*Marshall et al., 2016a*), and samples were submitted to BGI for sequencing.

Sequencing libraries were prepared at BGI Genomics following a ChIP-seq workflow. DNA fragments were first end-repaired and dA-tailed using End Repair and A-Tailing enzyme. Adaptors were then ligated for sequencing and ligated DNA purified using AMPure beads. DNA was then PCR amplified with BGI primers for eight cycles and PCR purified with AMPure beads. DNA was then homogenized, circularized, digested, and again purified. DNA was then prepared into proprietary DNA nanoballs (DNB) for sequencing on a BGISEQ-500 platform with 50 bp read length and 20 M clean reads.

## Quantification and statistical analyses

### Statistical analyses

For MB morphological analyses, results are presented as bar plots for which percentage of brains with MB lobe growth and/or guidance defects are calculated. For these analysis, N = number of brains examined. 'Growth defects' were defined by an overgrown, stunted, or absent lobe, and 'guidance defects' were defined by a full or partially misprojected lobe. In the case where both defect types were observed in a single brain, the defect was categorized as a 'growth and guidance defect'. All MB statistical analysis was performed using GraphPad Prism 8.4 (GraphPad Software, Inc, CA, USA). A 2 × N contingency table was used when comparing MB defects of more than two genotypes, where N = number of genotypes, and significance was determined using a chi-square test with either a Yates' or Bonferroni correction with * p<0.05, ** p<0.01, *** p<0.001, and **** p<0.0001.

GO enrichment analysis was done using PANTHER Overrepresentation testing (http://geneontology.org; *Mi et al., 2013*) with a Fisher's exact test (FDR < 0.05). Annotation version and release date: GO Ontology database DOI: 10.5281/zenodo.3873405, released 2020-06-01.

For targeted DamID analyses, sequencing data were aligned to release six of the *Drosophila melanogaster* genome and processed using damidseq_pipeline as previously described (*Marshall and Brand, 2017*; *Marshall et al., 2016a*; *Marshall and Brand, 2015b*). RNA Pol II occupancy over gene bodies was calculated via polii.gene.call (*Marshall et al., 2016a*; *Marshall et al., 2013*). DEGs were called via the NOIseq R package (*Tarazona et al., 2011*); briefly, RNA pol II gene occupancy scores were scaled, and inverse log values used as input to NOIseq with parameters of upper quantile normalization and biological replicates. DEGs were called with a q value of 0.95.

For CATaDa analyses, Dam-only BAM files generated via damidseq_pipeline were converted to 75 nt bins via bam2coverage, and replicates averaged. Peaks were called separately on the wild-type and mutant conditions via find_peaks (*Marshall et al., 2016a*; *Marshall et al., 2016b*) with a minimum quantile of 0.95, before combining and merging peaks with BEDTools (*Quinlan and Hall, 2010*), and plotting heat maps with SeqPlots (*Stempor and Ahringer, 2016*). Average Dam occupancy values over peaks for each biological replicate were determined via polii.gene.call and

differential occupancy called via NOIseq as with RNA Pol II occupancy above. Peaks called as significant were associated with the nearest gene promoter via peaks2genes (*Marshall et al., 2016a*).

For Dam-KDM5 analyses, after converting to bedgraphs via damidseq_pipeline as above, peaks were called using find_peaks (using the parameters fdr = 0.01, min_quant = 0.9) (*Marshall et al., 2016a*) on the averaged replicates, and genes overlapping peaks identified using peaks2genes (*Marshall et al., 2016a*).

## Acknowledgements

We thank all the frontline and essential workers who worked tirelessly to protect and assist others during the COVID-19 pandemic. We additionally thank Nicholas Baker, Hannes Bülow, Andreas Jenny, Bernice Morrow, Anna Francesconi, and all members of the Secombe Lab for their feedback and edits on the manuscript. We appreciate the confocal microscope training and assistance provided to us by Hillary Guzik and members of the Einstein Analytical Imaging Facility (AIF). We thank Gilbert Henry and Andrea Brand for their generous donations of fly stocks and reagents. Stocks obtained from the Bloomington *Drosophila* Stock Center (NIH P40OD018537) were also used in this study. The 1D4 and 9.4A monoclonal antibodies were obtained from the Developmental Studies Hybridoma Bank, created by the NICHD of the NIH and maintained at the University of Iowa. We are additionally grateful to the NIH Special Instrument Grant S10OD023591 and the Cancer Center Support Grant P30CA013330. This research was supported by the NIH Ruth L Kirschstein National Research Service Award F31NS110278, the Einstein MSTP Training Grant T32GM007288, and the Junior Investigator in Neuroscience Research Award (JINRA) from the Dominick P Purpura Department of Neuroscience to HAMH, NIH R01GM112783 and support from the Irma T Hirschl Trust to JS, and NHMRC grants APP1128784 and APP1185220 to OJM.

## Additional information

### Funding

| Funder | Grant reference number | Author |
|---|---|---|
| National Institutes of Health | R01GM112783 | Julie Secombe |
| National Health and Medical Research Council | APP1128784 | Owen J Marshall |
| National Health and Medical Research Council | APP1185220 | Owen J Marshall |
| National Institutes of Health | F31NS110278 | Hayden AM Hatch |
| National Institutes of Health | T32GM007288 | Hayden AM Hatch |
| Irma T. Hirschl Trust | | Julie Secombe |

The funders had no role in study design, data collection and interpretation, or the decision to submit the work for publication.

### Author contributions

Hayden AM Hatch, Conceptualization, Data curation, Formal analysis, Funding acquisition, Investigation, Methodology, Writing - original draft, Writing - review and editing; Helen M Belalcazar, Resources, Validation, Writing - review and editing; Owen J Marshall, Conceptualization, Formal analysis, Supervision, Funding acquisition, Investigation, Writing - review and editing; Julie Secombe, Conceptualization, Formal analysis, Funding acquisition, Investigation, Project administration, Writing - review and editing

### Author ORCIDs

Hayden AM Hatch (ID) https://orcid.org/0000-0001-5922-7291
Julie Secombe (ID) https://orcid.org/0000-0002-5826-7547

Decision letter and Author response
Decision letter https://doi.org/10.7554/eLife.63886.sa1
Author response https://doi.org/10.7554/eLife.63886.sa2

## Additional files

### Supplementary files

- Supplementary file 1. DEGs from $kdm5^{140}$ TaDa. Related to *Figure 5*.

- Supplementary file 2. GO categories from $kdm5^{140}$ TaDa. Related to *Figure 5*.

- Supplementary file 3. DEGs from $kdm5$ shRNA TaDa. Related to *Figure 5*.

- Supplementary file 4. GO categories from $kdm5$ shRNA TaDa. Related to *Figure 5*.

- Supplementary file 5. KDM5-regulated genes that are direct Dam-KDM5 targets. Related to *Figures 6* and *7*.

- Supplementary file 6. KDM5-regulated genes that are direct Dam-Pros targets. Related to *Figure 7*.

- Transparent reporting form

### Data availability

TaDa data have been deposited in GEO under the accession codes GSE156010 and GSE166116.

The following datasets were generated:

| Author(s) | Year | Dataset title | Dataset URL | Database and Identifier |
|---|---|---|---|---|
| Hatch HAM, Belalcazar HM, Marshall OJ, Secombe J | 2021 | Targeted DamID analyses of neural progenitor cells and immature neurons of kdm5 [140] and kdm5 shRNA Drosophila larvae and pupae. | https://www.ncbi.nlm. nih.gov/geo/query/acc. cgi?acc=GSE156010 | NCBI Gene Expression Omnibus, GSE156010 |
| Hatch HAM, Belalcazar HM, Marshall OJ, Secombe J | 2021 | Genome-wide binding profiles of KDM5 in Drosophila GMCs and immature neurons | https://www.ncbi.nlm. nih.gov/geo/query/acc. cgi?acc=GSE166116 | NCBI Gene Expression Omnibus, GSE166116 |

The following previously published datasets were used:

| Author(s) | Year | Dataset title | Dataset URL | Database and Identifier |
|---|---|---|---|---|
| Liu X, Shen J, Xie L, Wei Z, Wong C, Li Y, Zheng X, Li P, Song Y | 2020 | Genome-wide binding profiles of HP1a and Prospero in *Drosophila* central brain neural precursors and neurons | https://www.ncbi.nlm. nih.gov/geo/query/acc. cgi?acc=GSE136413 | NCBI Gene Expression Omnibus, GSE136413 |
| Drelon C, Belalcazar B, Secombe J | 2018 | RNA-seq analysis of kdm5 null mutant wing discs | https://www.ncbi.nlm. nih.gov/geo/query/acc. cgi?acc=GSE109201 | NCBI Gene Expression Omnibus, GSE109201 |

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

# Appendix 1

**Appendix 1—key resources table**

| Reagent type (species) or resource | Designation | Source or reference | Identifiers | Additional information |
|---|---|---|---|---|
| Antibody | Mouse monoclonal anti-brp | DSHB | Cat# nc82; RRID:AB_2314866 | IF (1:50) |
| Antibody | Rabbit monoclonal anti-HA-Tag | Cell Signaling Technology | Cat# 3724; RRID:AB_1549585 | IF (1:100) |
| Antibody | Mouse monoclonal anti-HA-Tag | Cell Signaling Technology | Cat# 2367; RRID:AB_10691311 | IF (1:100) WB (1:1000) |
| Antibody | Rat monoclonal anti-Deadpan | Abcam | Cat# ab195173; RRID:AB_2687586 | IF (1:100) |
| Antibody | Goat polyclonal anti-mouse Alexa-488 | Thermo Fisher Scientific | Cat# A32723; RRID:AB_2633275 | IF (1:500) |
| Antibody | Goat polyclonal anti-mouse Alexa-568 | Thermo Fisher Scientific | Cat# A11004; RRID:AB_2534072 | IF (1:500) |
| Antibody | Goat polyclonal anti-rabbit Alexa-488 | Thermo Fisher Scientific | Cat# A11034; RRID:AB_2576217 | IF (1:500) |
| Antibody | Goat polyclonal anti-rabbit Alexa-568 | Thermo Fisher Scientific | Cat# A11004, RRID:AB_2534072 | IF (1:500) |
| Antibody | Goat polyclonal anti-rat Alexa-568 | Thermo Fisher Scientific | Cat# A11077; RRID:AB_2534121 | IF (1:500) |
| Antibody | Mouse monoclonal anti-Fas2 | DSHB | Cat# 1D4; RRID:AB_528235 | IF (1:25) |
| Antibody | Rabbit polyclonal anti-KDM5 | Secombe Lab; *Secombe et al., 2007* | RRID:AB_2569502 | WB (1:1000) |
| Antibody | IRDye 680RD donkey monoclonal anti-mouse IgG secondary antibody | LI-COR Biosciences | LI-COR Biosciences Cat# 925-68072, RRID:AB_2814912 | WB (1:8000) |
| Antibody | IRDye 800CW donkey monoclonal anti-rabbit IgG secondary antibody | LI-COR Biosciences | LI-COR Biosciences Cat# 926-32213, RRID:AB_621848 | WB (1:8000) |

*Continued on next page*

*Appendix 1—key resources table continued*

| Reagent type (species) or resource | Designation | Source or reference | Identifiers | Additional information |
|---|---|---|---|---|
| Cell line (*Escherichia coli*) | NEB 5-alpha competent *E. coli* | New England BioLabs | Cat# C2987 | |
| Commercial assay or kit | Clontech CloneAmp HiFi PCR Premix | Clontech | Cat# 639298 | |
| Commercial assay or kit | Advantage 2 cDNA polymerase | Clontech | Cat# 639201 | |
| Commercial assay or kit | Agencourt AMPure XP Beads | Beckman Coulter | Cat# A63880 | |
| Commercial assay or kit | Takara In-Fusion HD Cloning Plus | Takara | Cat# 638909 | |
| Commercial assay or kit | Quick Ligation Kit | New England BioLabs | Cat# M2200S | |
| Commercial assay or kit | Zymo Quick-DNA miniprep plus | Zymo Research | Cat# D4069 | |
| Commercial assay or kit | Macherey-Nagel NucleoSpin Gel and PCR Clean-up Kit | Takara | Cat# 740609.250 | |
| Commercial assay or kit | Qubit dsDNA HS Assay Kit | Invitrogen | Cat# Q32851 | |
| Genetic reagent (*Drosophila melanogaster*) | *Drosophila: kdm5:3xHA* | This study | N/A | Endogenous kdm5:HA strain (*Figures 2–4*). Available from lead contact. |
| Genetic reagent (*D. melanogaster*) | *Drosophila: OK107-Gal4* | Bloomington Drosophila Stock Center | RRID:BDSC_854 | |
| Genetic reagent (*D. melanogaster*) | *Drosophila: 5XUAS-unc84::2XGFP* | Janelia Research | | Campus; *Henry et al., 2012* |
| N/A | | | | |
| Genetic reagent (*D. melanogaster*) | *Drosophila: wor-Gal4* | Bloomington Drosophila Stock Center | RRID:BDSC_56553 | |
| Genetic reagent (*D. melanogaster*) | *Drosophila: UAS-kdm5RNAI* | Bloomington Drosophila Stock Center | RRID:BDSC_35706 | |
| Genetic reagent (*D. melanogaster*) | *Drosophila: insc-Gal4* | Bloomington Drosophila Stock Center | RRID:BDSC_8751 | |
| Genetic reagent (*D. melanogaster*) | *Drosophila: c708a-Gal4* | Bloomington Drosophila Stock Center | RRID:BDSC_50743 | |
| Genetic reagent (*D. melanogaster*) | *Drosophila: c305a-Gal4* | Bloomington Drosophila Stock Center | RRID:BDSC_30829 | |
| Genetic reagent (*D. melanogaster*) | *Drosophila: H24-Gal4* | Bloomington Drosophila Stock Center | RRID:BDSC_51632 | |
| Genetic reagent (*D. melanogaster*) | *Drosophila: 201Y-Gal4* | Bloomington Drosophila Stock Center | RRID:BDSC_4440 | |

*Continued on next page*

*Appendix 1—key resources table continued*

| Reagent type (species) or resource | Designation | Source or reference | Identifiers | Additional information |
|---|---|---|---|---|
| Genetic reagent (*D. melanogaster*) | *Drosophila: UAS-Dcr-2* | Bloomington Drosophila Stock Center | RRID:BDSC_24650 | |
| Genetic reagent (*D. melanogaster*) | *Drosophila: GMR71C09-GAL4* | Bloomington Drosophila Stock Center | RRID:BDSC_39575 | |
| Genetic reagent (*D. melanogaster*) | *Drosophila: 20XUAS-IVS-CsChrimson.mVenus* | Bloomington Drosophila Stock Center | RRID:BDSC_55136 | |
| Genetic reagent (*D. melanogaster*) | *Drosophila: kdm5$^{NP4707}$* | Kyoto Stock Center; *Hayashi et al., 2002*; stock# 104754 | | |
| Genetic reagent (*D. melanogaster*) | *Drosophila: kdm5$^{140}$* | Secombe Lab; *Drelon et al., 2018* | | |
| Genetic reagent (*D. melanogaster*) | *Drosophila: UASt-kdm5* | Secombe Lab; *Secombe et al., 2007* | | |
| Genetic reagent (*D. melanogaster*) | *Drosophila: UAS-dMycRNAi* | Vienna Drosophila Resource Center | Stock# KK106066 | |
| Genetic reagent (*D. melanogaster*) | *Drosophila: UAS-prosRNAi* | Bloomington Drosophila Stock Center | RRID:BDSC_42538 | |
| Genetic reagent (*D. melanogaster*) | *Drosophila: tubP-Gal80$^{ts}$* | Bloomington Drosophila Stock Center | RRID:BDSC_7019 | |
| Genetic reagent (*D. melanogaster*) | *Drosophila: UAS-LT3-NDam* | Brand Lab; *Southall et al., 2013* | | |
| Genetic reagent (*D. melanogaster*) | *Drosophila: UAS-LT3-NDam-RpII215* | Brand Lab; *Southall et al., 2013* | | |
| Genetic reagent (*D. melanogaster*) | *Drosophila: w$^{1118}$* | Bloomington Drosophila Stock Center | RRID:BDSC_5905 | |
| Genetic reagent (*D. melanogaster*) | *Drosophila: UAS-LT3-NDam-kdm5* | This study | N/A | Used for KDM5 TaDa (*Figures 6* and *7*). Available from lead contact. |
| Sequence-based reagent | AdRt | *Vogel et al., 2007* | PCR primers | CTAATACGACTCACTA TAGGGCA GCGTGGTCGCGGCCGAGGA |
| Sequence-based reagent | AdRb | *Vogel et al., 2007* | PCR primers | TCCTCGGCCG |
| Sequence-based reagent | DamID_PCR | *Vogel et al., 2007* | PCR primers | GGTCGCGGCCGAGGATC |
| Sequence-based reagent | scram_shRNA | This study | PCR primers | GGATAATAGAATAGTTATATT CAAGCATATTCTATTATCC |
| Sequence-based reagent | Fw DsRed_KDM5_AarI | This study | PCR primers | tatagtgtcttcggggccgaCAGGAG CTGTGGCGCATTCTAGAAAC |
| Sequence-based reagent | PAM Rv | This study | PCR primers | AATCTGGAACATCGTATGGGT ACTGCGGCCGCGCTCGCGC |

*Continued on next page*

*Appendix 1—key resources table continued*

| Reagent type (species) or resource | Designation | Source or reference | Identifiers | Additional information |
|---|---|---|---|---|
| Sequence-based reagent | PAM Fw | This study | PCR primers | AGCAGCGGGCGGTGCAATCGG CGCGAGCGCGGCCGCAGTA |
| Sequence-based reagent | Rv DsRed_KDM5_AarI | This study | PCR primers | gattatctttctagggttaaAGGAAAAAGTC AAATAAAACGTAAGAAAAC TTTGC |
| Sequence-based reagent | Fw DsRed_KDM5_SapI | This study | PCR primers | gactatctttctagggttaaTCAAA GGCGAAGGCGACTCT |
| Sequence-based reagent | Rv DsRed_KDM5_SapI | This study | PCR primers | atatggtcttcttttcccggAACATGTTC CTCTTTTAAGGTGCTCTTT |
| Sequence-based reagent | Dam-kdm5_NotI-Fw | This study | PCR primers | cgcagatctgcggccgATGTCC GCCAAAACTGAGG |
| Sequence-based reagent | Dam-kdm5_XbaI-Rv | This study | PCR primers | acaaagatcctctagCTACCG CGCCGATTGCAC |
| Recombinant DNA reagent | pU6-BbsI-chiRNA | Addgene; *Gratz et al., 2013* | RRID: Addgene_ 45946 | |
| Recombinant DNA reagent | pValium20 | Drosophila Genomics Resource Center; *Ni et al., 2009* | DGRC# 1467 | |
| Recombinant DNA reagent | pHD-ScarlessDsRed | Drosophila Genomics Resource Center | DGRC# 1364 | |
| Software, algorithm | Fiji | https://fiji.sc/ | RRID:SCR_ 002285 | |
| Software, algorithm | Prism 6 | GraphPad | RRID:SCR_ 002798 | |
| Software, algorithm | Cytoscape | https://cytoscape. org/ | RRID:SCR_ 003032 | |
| Software, algorithm | Gene Ontology | http://www. geneontology.org | RRID:SCR_ 002811 | |
| Software, algorithm | R 3.5.1 | The R Foundation | RRID:SCR_ 001905 | |
| Software, algorithm | ggplot2 (R package) | CRAN | RRID:SCR_ 014601 | |
| Software, algorithm | damidseq_pipeline | *Marshall and Brand, 2015a* | http:// owenjm. github. io/ damidseq_ pipeline/ | |
| Software, algorithm | find_peaks | *Wolfram et al., 2012* | http:// github.com/ owenjm/ find_peaks | |
| Software, algorithm | bowtie2 | *Langmead and Salzberg, 2012* | http:// bowtie-bio. sourceforge. net /bowtie2/ index.shtml | |

*Continued on next page*

*Appendix 1—key resources table continued*

| Reagent type (species) or resource | Designation | Source or reference | Identifiers | Additional information |
|---|---|---|---|---|
| Software, algorithm | BioVenn | *Hulsen et al., 2008* | https://www.biovenn.nl/ | |
| Other | Vectashield Mounting Medium | Vector Labs | Cat# H-1000 | |
| Other | Normal donkey serum | Fisher Scientific | Cat# 50-413-367 | |
| Other | DAPI-Fluoromount-G | SouthernBiotech | Cat# OB010020 | |
| Other | T4 DNA ligase | New England BioLabs | Cat# M0202 | |
| Other | T4 polynucleotide kinase | New England BioLabs | Cat# M0201 | |
| Other | AarI | Thermo Scientific | Cat# ER1581 | |
| Other | SapI | New England BioLabs | Cat# R0569 | |
| Other | BpiI | Thermo Scientific | Cat# ER1011 | |
| Other | NotI | New England BioLabs | Cat# R0189 | |
| Other | PstI | New England BioLabs | Cat# R0140 | |
| Other | NheI | New England BioLabs | Cat# R0131 | |
| Other | EcoRI | New England BioLabs | Cat# R0101 | |
| Other | DpnI | New England BioLabs | Cat# R0176 | |
| Other | DpnII | New England BioLabs | Cat# R0543 | |
| Other | AlwI | New England BioLabs | Cat# R0513 | |

