## [Decision Letter]

**Acceptance summary:**

This study uses *Drosophila* to examine KDM5, histone H3K4 demethylase, that has been implicated in intellectual disabilities. The results show how KDM5 in the mushroom body is required for axonal development, independently of its histone demethylase activity. Insights revealed by this study include how KDM5 is involved in the growth and development of ganglion mother cells and immature neurons.

**Decision letter after peer review:**

Thank you for submitting your article "A KDM5-Prospero transcriptional axis functions during early neurodevelopment to regulate mushroom body formation" for consideration by *eLife*. Your article has been reviewed by three peer reviewers, and the evaluation has been overseen by a Reviewing Editor and K VijayRaghavan as the Senior Editor.

The reviewers have discussed the reviews with one another and the Reviewing Editor has drafted this decision to help you prepare a revised submission.

As the editors have judged that your manuscript is of interest, but as described below that additional experiments or substantial rewriting are required before it is published, we would like to draw your attention to changes in our revision policy that we have made in response to COVID-19 (https://elifesciences.org/articles/57162). First, because many researchers have temporarily lost access to the labs, we will give authors as much time as they need to submit revised manuscripts. We are also offering, if you choose, to post the manuscript to bioRxiv (if it is not already there) along with this decision letter and a formal designation that the manuscript is "in revision at *eLife*". Please let us know if you would like to pursue this option. (If your work is more suitable for medRxiv, you will need to post the preprint yourself, as the mechanisms for us to do so are still in development.)

Summary:

Overall the reviews of your manuscript were mixed, with two reviewers very positive, and one reviewer with several important comments that need to be addressed. Nevertheless, the consensus was that the manuscript is sufficiently interesting, well written, and novel enough to warrant the consideration of a revised version.

Reviewer #3 in particular had several concerns that would either require either more experiments, or the re-focusing of the central story of the manuscript. I will leave it to the authors to decide which option to take.

Essential revisions:

The suggested extra experiments are:

1) Perform their mutant analysis with entirely different genetics: Making these lines will take time – although maybe since the lab has studied the role of KDM5 in many biological settings they have these lines already?

2) Perform their Tada and behaviour analyses with more specific Gal4 lines, if developmentally expressed, MB-specific Gal4 lines exist.

3) Perform more genetic epistatic experiments with pros.

If the authors prefer to re-focus their story, however, this is the specific suggestion from reviewer #3:

"In my opinion, what's interesting is that different domains of KDM5 have different functions. In their 2018 paper (Zamurrad et al., 2018) they generated fly lines carrying mutations linked with ID in humans. These lines showed no neuroanatomical defects in the mushroom body but had defects in short and long term memory.

In this paper that we're discussing, the authors focus on a mutation that affects a different part of the gene. This, they find, causes severe neuroanatomical defects in the MB (probably elsewhere too), likely through its interaction with prosperso, a potent transcriptional regulator. This is interesting! I would urge the authors to make this the centerpiece of their story:

1) Thoroughly describe the cellular mechanism of the phenotype.

2) Thoroughly establish the KDM5>pros genetic link.

3) Describe the transcriptional programme that is affected.

If they chose to do this, I don't think that they need to focus on the mushroom body – both pros and KDM5 are ubiquitously expressed. This way, I also think that the behaviour experiments are not necessary. And finally, this way, the authors need not revisit their Tada experiments. They could focus only on getting the genetics of the cellular phenotypes and the pros interaction going and on re-focussing the text."

The specific reviews are appended:

Reviewer #1:

In this paper, Hatch et al. describe how KDM5 functions within the developing adult *Drosophila* brain to control normal formation of the mushroom body (MB). They show that loss of KDM5 functions within specific neural progenitor cells leads to defects in neuronal guidance and growth within the MB, and leads to disrupted cognitive function. Using a nice combination of whole-genome expression profiling approaches, they identify the gene targets regulated by KDM5 with neural progenitor cells. In particular they identify Prospero as a key downstream mediator of KDM5 function in the MB.

Overall, I liked this paper a lot. The experiments are all well performed, the paper is well written, and, most importantly, the discoveries should be of interest to a broad group of developmental and behavioural neurobiologists.

Reviewer #2:

The work describes the first cell-type-specific analysis of histone H3K4 demethylase KDM5 in fly brain development. The experiments and analyses appear to be well designed and executed. The data support most of the claims. The authors used the state-of-the-art genomic profiling assays for pol II binding and chromatin accessibility. An important finding is the striking overlap of pro occupancy and KDM5-regulated genes, which is backed up by the robust genetic data demonstrating the functional interaction of the two genes. Mutations in multiple KDM5-family enzymes are responsible for human neurodevelopmental disorders. Thus, the work is an essential step towards understanding the pathophysiology of the conditions.

Reviewer #3:

In this manuscript the authors study the role gene histone demethylase gene, KDM5, in the development of the mushroom body. In vertebrates, loss of function of homologues of this gene have been shown to be involved in intellectual disabilities. The authors generate a tagged form of the protein to assess its expression pattern and find it to be expressed ubiquitously in the brain. They show that a loss of this gene results in dramatic defects in the mushroom body and use a combination of Gal4 lines to narrow down which cells it might be required in – stem cells, intermediate precursors and immature neurons. They profile these brains transcriptionally and identify targets that share >40% overlap with Pros target genes. They test a possible interaction KDM5 and pros through a transheterozygous interaction between the two genes.

The manuscript is generally well written and presented.

[Editors' note: further revisions were suggested prior to acceptance, as described below.]

Thank you for submitting your article "A KDM5-Prospero transcriptional axis functions during early neurodevelopment to regulate mushroom body formation" for consideration by *eLife*. Your article has been reviewed by three peer reviewers, and the evaluation has been overseen by a Reviewing Editor and K VijayRaghavan as the Senior Editor. The reviewers have opted to remain anonymous.

Essential Revisions:

Please see the revisions suggested by reviewer #3. Upon consultation, it was decided that none of these revisions are necessarily essential, but if the authors could make some efforts to 1) increase cellular resolution analysis, and 2) strengthen the KDM5-pros link, in the manner described below. If this is not feasible for any reason please let us know.

Reviewer #1:

This paper uses a nice combination of genetics and whole-genome investigation of transcription to reveal a kdm5-pros transcriptional module that is important for regulating neuronal gen expression and for controlling development of the mushroom body. Given that both kdm5 and pros are conserved, these findings may be important for our understanding of neuronal gene expression in other contexts.

In this revised manuscript, the authors provide new data that strengthens the original paper. I particularly liked the new kdm5 Dam ID experiments showing kdm5 localization at the pros gene – these results, along with the original genetic epistasis experiments, emphasize an important kdm5-pros transcriptional network in the control of MB development.

The authors have satisfactorily addressed all my original comments and I am happy to accept the paper for publication in *eLife*. Congratulations to the authors on a very nice piece of work

Reviewer #2:

The work describes the first cell-type-specific analysis of histone H3K4 demethylase KDM5 in fly brain development. The experiments and analyses appear to be well designed and executed. The data support most of the claims. The authors used the state-of-the-art genomic profiling assays for pol II binding and chromatin accessibility. An important finding is the striking overlap of pro occupancy and KDM5-regulated genes, which is backed up by the robust genetic data demonstrating the functional interaction of the two genes. Furthermore, the authors were able to show pro is a direct target gene KDM5C regulates in a specific neuron type of the fly brain. Mutations in multiple KDM5-family enzymes are responsible for human neurodevelopmental disorders. Thus, the work is an essential step towards understanding the pathophysiology of the conditions.

The authors addressed comments from the reviewers well. It is nice to see that KDM5C directly regulates the expression of pro rather than the two factors work in concert.

Reviewer #3:

In this manuscript the authors study the role of a histone demethylase gene, KDM5, in the development of the mushroom body. In vertebrates, loss of function of homologues of this gene have been shown to be involved in intellectual disabilities. The authors generate a tagged form of the protein to assess its expression pattern and find it to be expressed ubiquitously in the brain. They show that a loss of this gene results in dramatic defects in the mushroom body and use a combination of Gal4 lines to narrow down which cells it might be required in – stem cells, intermediate precursors and immature neurons. They profile these brains transcriptionally, as well profile KDM5 binding in these brains. They find that KDM5 binds to the pros locus, whose transcript is down regulated in the absence of KDM5. Interestingly they show that this is not due to KDM5's role in chromatin regulation. They find a significant overlap in the targets of KDM5 and pros and postulate that the two act together in growth and patterning. They test a possible interaction between KDM5 and pros through a transheterozygous interactions between the two genes in MB patterning.

While interesting, I have some concerns that I am listing below.

My major concern with this manuscript is the resolution of analysis. The authors analyze mutant phenotypes and assign cell-specific requirements based on Gal4 drivers and their previously ascribed cell specificities. A more thorough approach would have been to assess the requirement of KDM5 in clonal analysis that irrefutably manipulates the function of KDM5 in these specific cell types and then assess their effects at a cellular level.

The authors profile the transcriptome of KDM5 depleted GMCs and neurons using the broadly expressed R71C09-Gal4. Similarly, they assay for KDM5 binding in the R71C09-Gal4 domain. As the knockdown of KDM5 is also dependent on the Gal4 (or is a whole animal mutant), in my view, the authors are studying these regulatory processes and interactions in all NBs and GMCs. They however interpret these data as being specific to the mushroom body due to the timing of the experiments. I do not understand the reasoning behind this interpretation.

I find the data interesting. I believe the authors have uncovered a general process by which neurons mature and form connections as both pros and KDM5 are ubiquitously expressed. This will be of interest – as will the datasets that have been generated in this study – to the field in general.

I think this revised manuscript is much better without the behaviour. However, two of my earlier concerns still remain unaddressed. I'll reiterate them here:

1) I would recommend improved cellular resolution of their analysis. This could be done by performing clonal analysis to show the specific cell type in which KDM5 is required. The authors could use the MB as an illustrative cell-type, but I don't think it necessary – KDM5 is ubiquitously expressed and their accompanying TaDa analysis has been done with the broadly expressed R71C09-Gal4. The authors could take this decision based on whatever genetics is easier done. Such clonal analysis will also give the authors far better cellular resolution of their phenotypes.

2) Strengthen their KDM5-pros link. In this revised manuscript, the authors show that KDM5 binds pros, but the binding seems barely above background (was it picked up in a peak call?). They also say that no physical interaction was picked up between the two in IP experiments. So, if this is central to their story, I would recommend they build this axis with more convincing data. For example, if KDM5 acts through pros, is pros protein downregulated in the KDM5 null? (There are very good pros antibodies out there.). Can Pros rescue KDM5 mutant phenotype?

---

## [Author Response]

Essential revisions:The suggested extra experiments are:1) Perform their mutant analysis with entirely different genetics: Making these lines will take time – although maybe since the lab has studied the role of KDM5 in many biological settings they have these lines already?2) Perform their Tada and behaviour analyses with more specific Gal4 lines, if developmentally expressed, MB-specific Gal4 lines exist.3) Perform more genetic epistatic experiments with pros.If the authors prefer to re-focus their story, however, this is the specific suggestion from reviewer #3:"In my opinion, what's interesting is that different domains of KDM5 have different functions. In their 2018 paper (Zamurrad et al., 2018) they generated fly lines carrying mutations linked with ID in humans. These lines showed no neuroanatomical defects in the mushroom body but had defects in short and long term memory.In this paper that we're discussing, the authors focus on a mutation that affects a different part of the gene. This, they find, causes severe neuroanatomical defects in the MB (probably elsewhere too), likely through its interaction with prosperso, a potent transcriptional regulator. This is interesting! I would urge the authors to make this the centerpiece of their story:1) Thoroughly describe the cellular mechanism of the phenotype.2) Thoroughly establish the KDM5>pros genetic link.3) Describe the transcriptional programme that is affected.If they chose to do this, I don't think that they need to focus on the mushroom body – both pros and KDM5 are ubiquitously expressed. This way, I also think that the behaviour experiments are not necessary. And finally, this way, the authors need not revisit their Tada experiments. They could focus only on getting the genetics of the cellular phenotypes and the pros interaction going and on re-focussing the text."

In response to editor and reviewer suggestions, we have included data which significantly strengthen our model linking KDM5 and Prospero (Figures 6,7; Supplementary files 5,6). Specifically, we generated a fly strain expressing a UAS-dam-kdm5 transgene under control of the R71C09-Gal4 driver. This allowed us to assess direct genetic targets of KDM5 within immature neurons and GMCs during a similar developmental window as for our Pol II TaDa datasets. Not only does this represent the first cell-specific analysis of KDM5 occupancy, but it has allowed us to demonstrate that KDM5 directly binds to pros at transcriptional start sites to regulate its expression (Figure 6B-D; Figure 7A-C). This additional data provide clear support for our model that a KDM5-Pros transcriptional axis is critically important for mushroom body development.

[Editors' note: further revisions were suggested prior to acceptance, as described below.]

Essential Revisions:Please see the revisions suggested by reviewer #3. Upon consultation, it was decided that none of these revisions are necessarily essential, but if the authors could make some efforts to 1) increase cellular resolution analysis, and 2) strengthen the KDM5-pros link, in the manner described below. If this is not feasible for any reason please let us know.Reviewer #3:[…]I think this revised manuscript is much better without the behaviour. However, two of my earlier concerns still remain unaddressed. I'll reiterate them here:1) I would recommend improved cellular resolution of their analysis. This could be done by performing clonal analysis to show the specific cell type in which KDM5 is required. The authors could use the MB as an illustrative cell-type, but I don't think it necessary – KDM5 is ubiquitously expressed and their accompanying TaDa analysis has been done with the broadly expressed R71C09-Gal4. The authors could take this decision based on whatever genetics is easier done. Such clonal analysis will also give the authors far better cellular resolution of their phenotypes.

It is unclear from reviewer 3’s comment what significant new information examining the effects of *kdm5* loss at a single cell level would provide to our investigation. Our current cell type-specific knockdown data already convincingly demonstrate that KDM5 is required within ganglion mother cells (GMCs) and immature neurons for proper axonal growth and guidance. While the 71C09-Gal4 driver is expressed in many GMCs within the brain, we chose to focus our analyses on the Mushroom Body due to its highly characterized morphology and links to cognitive function. We do, however, note in text that depletion of KDM5 using this Gal4 driver could lead to defects within other neuronal structures, the characterization of which are outside the scope of this manuscript.

2) Strengthen their KDM5-pros link. In this revised manuscript, the authors show that KDM5 binds pros, but the binding seems barely above background (was it picked up in a peak call?). They also say that no physical interaction was picked up between the two in IP experiments. So, if this is central to their story, I would recommend they build this axis with more convincing data. For example, if KDM5 acts through pros, is pros protein downregulated in the KDM5 null? (There are very good pros antibodies out there.). Can Pros rescue KDM5 mutant phenotype?

Regarding the Dam:KDM5 TaDa binding data, it is possible that reviewer 3 is more familiar with looking at ChIP-seq genome browser tracks and not those generated by TaDa analyses. As described in detail in the Materials and methods section, Dam:KDM5 binding at the *pros* gene was significant when compared to expression of Dam alone (using a 1% FDR cutoff and six biological replicates). To further clarify this matter, we now show the two peaks called within the *pros* gene in Figure 7, which have p(adj) values of 0.00052 and 0.00363, respectively. A file containing all significant called peaks, along with p(adj) values has been added to the referenced GEO submission.

Reviewer 3’s second concern relates to whether levels of Pros protein are affected by KDM5 depletion as assessed via IHC. We have performed IHC staining of pupal and adult brains using two independent Pros antibodies, one of which was obtained from Drs. Lily and Yuh-Nung Jan and the other from the Developmental Studies Hybridoma Bank. Although these antibodies work well in embryonic and larval brain tissue, they do not clearly detect Pros protein within pupal and adult tissue. The reason for this is unclear but may be related to the fact that these antibodies do not detect all Pros isoforms.

Reviewer 3’s third concern relates to whether overexpression of Pros can rescue the *kdm5^140^* mutant (or knockdown) phenotype. We obtained a *UAS-Pros* transgene from the Bloomington *Drosophila* Stock Center to carry out this experiment. Unfortunately, overexpression of Pros using *R71C09-Gal4* kills animals during early embryonic and larval development. Additionally, overexpression of Pros within Kenyon cells using *OK107-Gal4* results in complete ablation of the mushroom body. This toxicity prevented us from performing the suggested rescue experiments.